# FOURIERHANDFLOW: Neural 4D Hand Representation Using Fourier Query Flow

Jihyun Lee[1]    Junbong Jang[1]    Donghwan Kim[1]    Minhyuk Sung[1]    Tae-Kyun Kim[1,2]

[1]KAIST        [2]Imperial College London

{jyun.lee, junbongjang, kdoh2522, mhsung, kimtaekyun}@kaist.ac.kr

## Abstract

Recent 4D shape representations model continuous temporal evolution of implicit shapes by (1) learning query flows without leveraging shape and articulation priors or (2) decoding shape occupancies separately for each time value. Thus, they do not effectively capture implicit correspondences between articulated shapes or regularize jittery temporal deformations. In this work, we present FOURIER-HANDFLOW, which is a spatio-temporally continuous representation for human hands that combines a 3D occupancy field with articulation-aware query flows represented as *Fourier series*. Given an input RGB sequence, we aim to learn a fixed number of Fourier coefficients for each query flow to guarantee smooth and continuous temporal shape dynamics. To effectively model spatio-temporal deformations of *articulated* hands, we compose our 4D representation based on two types of Fourier query flow: (1) pose flow that models query dynamics influenced by hand articulation changes via implicit linear blend skinning and (2) shape flow that models query-wise displacement flow. In the experiments, our method achieves state-of-the-art results on video-based 4D reconstruction while being computationally more efficient than the existing 3D/4D implicit shape representations. We additionally show our results on motion inter- and extrapolation and texture transfer using the learned correspondences of implicit shapes. To the best of our knowledge, FOURIERHANDFLOW is the first neural 4D continuous hand representation learned from RGB videos. Our code is available at https://github.com/jyunlee/FourierHandFlow.

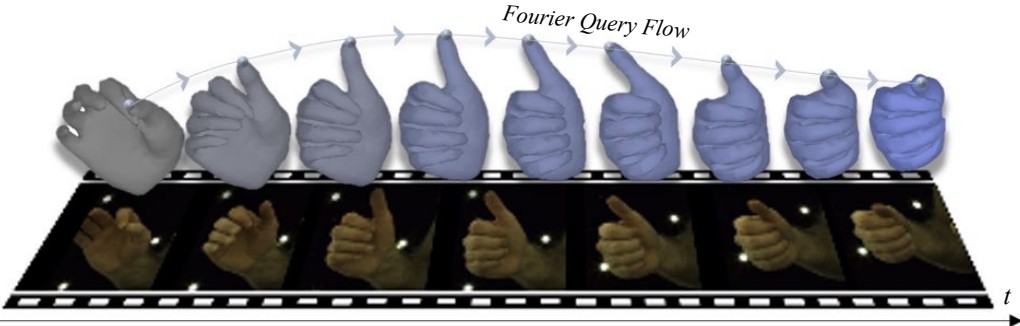

Figure 1: From monocular RGB sequence inputs, FOURIERHANDFLOW learns 4D hand shapes that are continuous in both space and time. It models temporal shape evolutions with query flows learned as a fixed number of coefficients for Fourier series to guarantee smooth temporal dynamics.

37th Conference on Neural Information Processing Systems (NeurIPS 2023).

# 1 Introduction

Neural implicit representations [35, 18, 16, 23, 30, 25, 24, 27] have achieved appealing results in modeling resolution-free 3D articulated shapes, such as human bodies and hands. Motivated by their success in 3D domain, several recent methods [14, 15, 29, 37, 39, 33, 37] have proposed to model time-varying implicit shapes that are continuous in both space and time. Pioneering work in this direction, Occupancy Flow [29], learns an occupancy field with free-form query flows (*i.e.* trajectories) over time using Neural ODE [5] to model continuous temporal deformations. However, it is prone to produce spatially over-smooth shapes (*e.g.* hands with fingers not separated) due to the lack of shape category and articulation priors. Following works on modeling human bodies [14, 15] address this issue by disentangling pose and shape-dependent deformations in latent space and decoding an occupancy field separately for each time value. However, they (1) cannot easily regularize abrupt or jittery motions, (2) are not computationally efficient, and (3) do not capture temporal correspondences between shapes – all due to occupancy decoding for each time sampling. Also, they are learned using a parametric model [21, 32] together with sparse geometry inputs (*e.g.* RGB-D or point clouds), thus it is non-trivial to adapt most of them to learn directly from RGB inputs alone.

In this paper, we aim to tackle a less-explored problem of learning a spatio-temporally continuous representation from monocular RGB sequences for human hands. To address the aforementioned limitations, we focus on learning a representation that exhibits the following desired properties:

1. *Continuous and smooth 4D reconstruction*: The learned temporal evolution of 3D implicit hands should be continuous and smooth (*i.e.* without temporal jitters or abrupt motions).

2. *Computational efficiency*: The representation should allow computationally efficient 4D reconstruction.

3. *Articulated shape modeling with correspondences*: The representation should effectively model *articulated* hand geometries while capturing implicit shape correspondences.

To this end, we present a novel pixel-aligned 4D hand representation, namely FOURIERHANDFLOW. To meet *continuous and smooth 4D reconstruction* and *computational efficiency*, we propose to combine a continuous 3D hand occupancy field with articulation-aware query flows represented as *Fourier series* along the temporal axis, which is parameterized by the coefficients learned from an input RGB sequence. In particular, we aim to learn coefficients for a fixed number of low-mid frequency terms to naturally guarantee smooth temporal evolution of 3D hands. This effectively imposes frequency-domain constraints on the learned motions to avoid jittery or abrupt temporal deformations, which are non-trivial to regularize using the existing 4D continuous representations [29, 15, 14]. Also, by representing a query flow as a sum of the continuous trigonometric basis functions, we can preserve temporal shape continuity while being more computationally efficient than the existing 4D continuous representations [29, 15] that require solving ODEs [5, 38] or occupancy decoding separately at each time sampling [15, 14]. In the experiments (Sec. 4.3), we also show that learning query flows in the frequency domain achieves better accuracy than that in the spatial domain.

For *articulated shape modeling with correspondences*, we compose our 4D hand representation based on two types of Fourier query flow: (1) pose flow and (2) shape flow. For the pose flow, we first estimate Fourier coefficients representing the flow of 3D hand joints, namely *joint flow*, to model the temporal change of hand articulation. For effective estimation of the joint flow, we take as input the noisy per-frame joint predictions obtained from an off-the-shelf hand pose estimator [19] and learn to denoise them in the frequency domain. The estimated joint flow is then propagated to each query point using linear blend skinning (LBS) to compute the per-query pose flow, for which we utilize a pre-trained implicit LBS weight field. Next, our shape flow is learned as the Fourier coefficients representing per-query *displacement* flow *w.r.t.* the previously estimated pose flow. Shape flow aims to model spatio-temporal deformations (*e.g.* soft-tissue or identity-dependent deformations) that cannot be expressed in the prior stage alone (*i.e.*, linear deformations *w.r.t.* the joint articulation changes). Our final query flow is obtained as a sum of the learned pose and shape flows. Unlike the existing 4D continuous human representations [15, 14] that disentangle pose and shape *in feature space* and decode 3D occupancies separately at each time step, our method captures correspondences between the implicit shapes by modeling query flows directly *in the target spatio-temporal space*.

To the best of our knowledge, FOURIERHANDFLOW is the first work proposed for spatio-temporally continuous hand representation learned from RGB videos. In the experiments, we validate the effec-

tiveness of FOURIERHANDFLOW on video-based 4D hand reconstruction using InterHand2.6M [26] dataset, where we achieve state-of-the-art results in comparison to the existing (1) image-based 3D hand shape reconstruction methods and (2) 4D implicit shape reconstruction methods modified to take RGB hand sequences as inputs. Also, our inference speed is about $30\times$ and $500\times$ faster than the existing state-of-the-art 3D and 4D implicit hand functions, respectively. We additionally show the effectiveness of our method on motion inter- and extrapolation and texture transfer using the learned correspondences of implicit shapes. Lastly, we examine the generalization ability of our method on unseen RGB2Hands [41] real images.

## 2  Related Work

**4D representation for implicit shapes.** Motivated by the success of 3D implicit representations in shape modeling [35, 18, 16, 23, 30, 25, 24, 27], which is resolution-free (*i.e.* continuous in space), several methods have been proposed for 4D representation that models dynamic implicit shapes [14, 15, 29, 37, 39, 33, 37]. These methods mainly combine the existing 3D implicit shape representation (*e.g.* occupancy field [23]) with an additional mechanism to model continuous temporal evolution of shapes. Occupancy Flow [29] models such temporal dynamics by estimating free-form query flows over time via Neural ODE [5]. Other works [39, 33, 37] decode an occupancy field conditioned on a time value using temporal-aware shape features. However, these methods often produce spatially over-smooth shapes due to the lack of shape category and articulation priors, which is addressed by the following methods [14, 15] proposed for 4D implicit human representation. 4D-CR [15] learns shape, initial state, and motion-disentangled latent features from training point cloud pairs of the same motion with different identities synthesized using SMPL [21] model. LoRD [14] learns 4D humans using local representations conditioned on SMPL shapes iteratively registered to point cloud or RGB-D inputs. While these methods show promising results, they are learned by leveraging a parametric model [21, 32] together with sparse geometry input observations (*e.g.* 2.5D or 3D point clouds). Also, they cannot easily regularize abrupt or jittery motions and are not computationally efficient due to the use of an ODE solver [15] and/or occupancy decoding separately for each time value [14, 15]. To address these limitations, we propose to learn articulation-aware query flows in the frequency domain to enable *smooth* and *efficient* temporal modeling of shapes.

**Hand shape reconstruction.** 3D hand shape reconstruction has been an active area of research. Most of existing methods model 3D hand shapes by predicting MANO [34] model parameters [45, 10, 1, 2, 11, 12, 47, 44] or vertex positions of a template hand mesh [19, 17, 20, 40] from an input observation (*e.g.* RGB, depth, sparse joint positions). However, their hand reconstructions are constrained to a discretized representation of shape, which usually is a low-resolution mesh with MANO topology ($|\mathcal{V}| = 778$). To address this issue, several recent methods [16, 18, 43, 6] have adopted implicit shape representation to model resolution-free hand shapes. These implicit representations are also shown to reconstruct hand shapes that are better aligned to input RGB observations [18, 9].

The most related domain to our work, *temporal-aware* 3D hand shape reconstruction from RGB, had been however limited due to the lack of available datasets. While there are datasets and methods proposed for temporal-aware hand pose (*i.e.,* sparse joints) estimation [46, 3, 8, 28, 3, 22], there had been no large-scale dataset that contains sequences of RGB observations with accurate dense shape annotations until the release of InterHand2.6M [26]. Thus, few existing methods had created synthetic RGB datasets [42, 41] or perform self-supervised learning [4] for a hand shape tracking model. However, they only show qualitative results of *per-frame* mesh estimation, where our goal is to learn 4D continuous hand shapes to allow arbitrary-resolution reconstruction and motion inter- and extrapolation. In this work, we train our 4D hand model on the single-hand and two-hand subsets of InterHand2.6M and compare our results to more recent state-of-the-art 3D hand reconstruction methods [18, 44, 19, 45] on InterHand2.6M and 4D implicit functions [29, 14] adapted to learn hand shapes from RGB.

**Shape modeling using frequency domain.** We briefly discuss the existing work on frequency-domain shape modeling, specifically focusing on deep learning-based methods for shape reconstruction and generation. Shen *et al.* [36] proposes to estimate Fourier-domain slices from input images to enable computationally efficient 3D shape reconstruction. Hui *et al.* [13] introduces a wavelet-domain diffusion model to allow diffusion-based generative modeling directly on implicit shape representation. However, these methods mainly aim to model non-articulated objects (*e.g.* chairs, airplanes). Fourier Occupancy Field [9], which is more related work to ours, proposes to represent

an occupancy field with Fourier series along the $z$-axis to enable efficient human reconstruction. Similar to [9], we adopt Fourier series (*cf.* discrete Fourier or wavelet transform) to preserve the continuity of our representation in the target spatio-temporal space. In contrast to [9] that learns Fourier coefficients to model static occupancy field along one spatial dimension, we learn Fourier series to model query flows over time to capture continuous 4D deformations of hands.

# 3 FOURIERHANDFLOW: Pixel-Aligned 4D Hands with Fourier Query Flow

Our work aims to learn a 4D continuous hand representation from a sequence of single-view RGB frames $(\mathbf{I}^t)_{t=1}^{T}$, where $\mathbf{I}^t \in \mathbb{R}^{W \times H \times 3}$ is an RGB frame of a size $W \times H$ observed at time $t$. To maintain consistency with the existing 4D continuous representations [14, 29, 15], our method takes as input a sub-sequence of $T = 17$ consecutive frames at once. In what follows, we first briefly explain our pre-trained occupancy and linear blend skinning weight fields (Sec. 3.1), which are prepared prior to learning our query flows. We then explain our Fourier query flow representation (Sec. 3.2) and 4D hand representation using Fourier query flows to enable articulation-aware shape modeling (Sec. 3.3).

## 3.1 Pre-Training Occupancy and Linear Blend Skinning Weight Fields

The main focus of our work is to learn query flows $\Phi(\cdot, \cdot) : \mathbb{R}^3 \times [0, T) \rightarrow \mathbb{R}^3$ that models a 3D trajectory of a query point $\mathbf{p} \in \mathbb{R}^3$ over the time span $t \in [0, T)$. To perform 4D hand reconstruction, we propagate a pre-trained hand occupancy field in the canonical coordinate system to the coordinate system at target time $t$ using the learned query flows – without requiring neural occupancy decoding separately for each $t$. To obtain our canonical occupancy field, we learn an occupancy function $o(\cdot) : \mathbb{R}^3 \rightarrow [0, 1]$ that maps a query point $\mathbf{p}$ in the canonical space to the occupancy probability in $[0, 1]$. Along with $o(\cdot)$, we also learn an implicit linear blend skinning (LBS) function $w(\cdot) : \mathbb{R}^3 \rightarrow [0, 1]^B$ that maps $\mathbf{p}$ to an LBS weight vector for $B$ hand bones, which will be later used to model query flows induced by the change of hand articulation (Sec 3.3). We learn both $o(\cdot)$ and $w(\cdot)$ prior to learning our query flows, for which we train a modified version of LEAP [25] network. For more details and backgrounds on LBS, please refer to the supplementary section.

Note that, while Occupancy Flow [29] takes a similar approach and propagates an occupancy field at $t = 0$ learned for each sequence separately, we propagate a canonical hand occupancy field that is shared among all sequences to allow modeling implicit correspondences between hand shapes from different sequences. Also, learning occupancy in the canonical system (*cf.* observation-specific system) is known to be more robust as addressed in the recent implicit functions [25, 16, 18].

## 3.2 Fourier Query Flow Representation

We now introduce our query flow representation based on Fourier series. The representation for query flow $\Phi(\cdot, \cdot)$ (1) should be easy to be learned for *smooth* query dynamics (*e.g.*, without jitters and abrupt motions) and (2) should be computationally efficient. However, it is difficult to achieve both merits using the existing 4D continuous representations [14, 15, 29] with an ODE solver [15, 29] or occupancy decoding for each time step [14, 15]. To address these limitations, we explore the Fourier-based representation for query flows. Specifically, we consider a set of the sine-cosine form of Fourier series $\{f^d(\cdot)\}_{d=x,y,z}$, where each $f^d(\cdot)$ represents the flow of an arbitrary query point in $d \in \{x, y, z\}$ dimension over the normalized time span $t \in [0, 1)$:

$$f^d(t) = \frac{\mathbf{a}_0^d}{2} + \sum_{n=1}^{N}(\mathbf{a}_n^d \cos(nt) + \mathbf{b}_n^d \sin(nt)), \quad \text{for } d \in \{x, y, z\}. \tag{1}$$

In Eq. (1), $\mathbf{a}^d \in \mathbb{R}^{N+1}$ and $\mathbf{b}^d \in \mathbb{R}^N$ are coefficients of cosine and sine basis functions defined in $d \in \{x, y, z\}$ dimension, respectively. $N$ denotes the number of each basis function, which would be theoretically an infinity to represent an arbitrary signal. In this work, as most of high-frequency hand motions are unnatural to occur in the real world, we aim to learn coefficients for a fixed number ($N = 6$) of basis functions. This eases the training of neural networks by constraining the number of Fourier coefficients to learn, while guaranteeing the reconstruction of smooth query dynamics along the temporal axis. Also, note that Eq. (1) naturally models a continuous query trajectory over time by

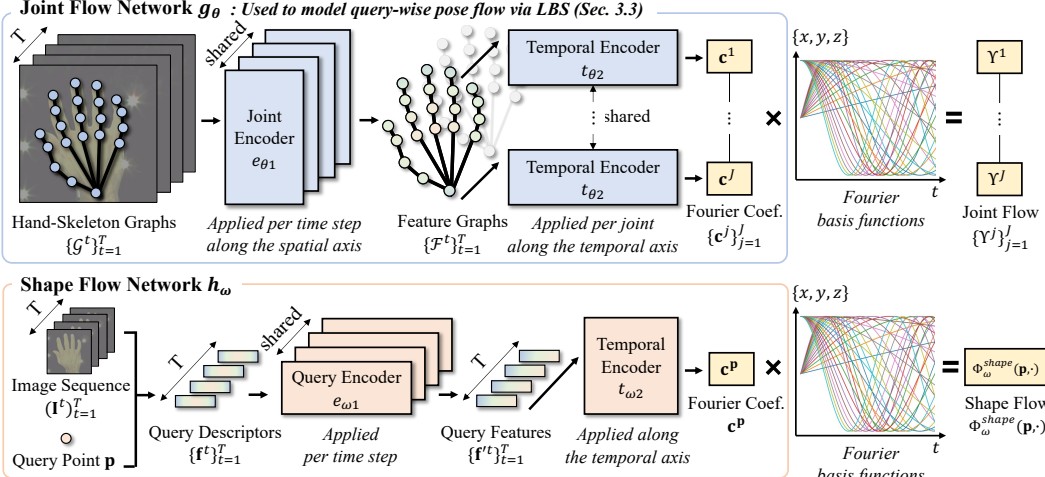

Figure 2: We model a query flow by estimating the joint flow (blue upper branch) – which is used to compute query-wise pose flow – and the shape flow (pink lower branch). Our joint flow network first predicts the Fourier coefficients for the flows of hand joint positions using the joint encoder and the temporal encoder. The predicted joint flow is propagated to a query point via implicit linear blend skinning to compute the pose flow (see Sec. 3.3). Our shape flow network then predicts the Fourier coefficients for the query-wise displacement flow $w.r.t.$ the estimated pose flow using the query encoder and the temporal encoder. Our final query flow is computed as a sum of the pose and shape flows.

representing it as a sum of trigonometric functions. Once the coefficients for the basis functions are estimated per input sequence, sampling query positions along the temporal axis also does not require additional computations (*e.g.*, solving ODEs [29] or neural occupancy decoding [14, 15]) other than evaluating Eq. (1) with a different value of $t$ at $O(1)$ complexity.

### 3.3 Pixel-Aligned 4D Hand Representation Using Fourier Query Flow

We introduce our pixel-aligned 4D hand representation using the Fourier query flow representation discussed in Sec. 3.2. To enable articulation-aware shape modeling, our representation is based on two types of Fourier query flow: (1) pose flow and (2) shape flow.

**Pose flow.** Our pose flow aims to model a query flow influenced by the change of hand articulation. In this stage, instead of directly estimating flow for a query point, we first predict the Fourier query flow for 3D hand joint positions, namely *joint flow*. The learned joint flow is then propagated to an arbitrary query point using a pre-trained LBS field (Sec. 3.1) to model query-wise pose flow.

Let $\mathbf{c} \in \mathbb{R}^{6N+3}$ be the concatenation of all Fourier coefficients (*i.e.*, $\{\mathbf{a}_n^d\}_{n=0}^N$ and $\{\mathbf{b}_n^d\}_{n=1}^N$ for $d \in \{x, y, z\}$) that determines the Fourier flow of one query point (Eq. (1)). In this stage, we aim to learn the flows of $J = 21$ hand joint positions parameterized by $\{\mathbf{c}^j\}_{j=1}^J$, where each $\mathbf{c}^j \in \mathbb{R}^{6N+3}$ denotes the Fourier coefficients of $j$-th joint flow. To effectively learn $\{\mathbf{c}^j\}_{j=1}^J$ from an input RGB sequence, we condition our prediction on the 3D joint positions obtained per frame using an off-the-shelf pose estimator $\Psi(\cdot)$. Our joint flow network $g_\theta(\cdot)$ learns the mapping from $(\mathbf{I}^t)_{t=1}^T$ to $\{\mathbf{c}^j\}_{j=1}^J$ as follows:

$$g_\theta((\mathbf{I}^t)_{t=1}^T) = t_{\theta_2}(e_{\theta_1}((\mathbf{I}^t)_{t=1}^T, \Psi((\mathbf{I}^t)_{t=1}^T))). \tag{2}$$

In Eq. (2) above, $g_\theta(\cdot)$ is composed of two sub-networks $e_{\theta_1}(\cdot, \cdot)$ and $t_{\theta_2}(\cdot)$ parameterized by the weights $\theta = \{\theta_1, \theta_2\}$. Our joint encoder $e_{\theta_1}(\cdot, \cdot) : \mathbb{R}^{T \times W \times H \times 3} \times \mathbb{R}^{T \times J \times 3} \to \mathbb{R}^{T \times J \times f}$ first extracts $f$-dimensional per-joint features given an input RGB sequence and initial joints predicted by $\Psi(\cdot)$. As shown in the upper branch of Fig. 2, we represent the initial joints as a hand-skeleton graph $\mathcal{G}^t$ for each time step, where each node feature is set as the concatenation of 3D positions and

pixel-aligned features [35] of the corresponding hand joint predicted by $\Psi(\cdot)$. Then, the joint encoder applies graph convolutions [7] to extract a feature graph $\mathcal{F}^t$ with the updated per-joint features. Our temporal encoder $t_{\theta_2}(\cdot) : \mathbb{R}^{T \times J \times f} \to \mathbb{R}^{J \times (6N+3)}$ then temporally aggregates the per-joint features using temporal convolutions. These temporal-aware joint features are concatenated with the Fourier coefficients of the initial per-frame joints[1] and fed to an MLP for the prediction of refined coefficients $\{\mathbf{c}^j\}_{j=1}^J$. Finally, the flow of $j$-th hand joint $\Upsilon^j(\cdot) : [0, 1) \to \mathbb{R}^3$ over the normalized time span $[0, 1)$ is modeled by Eq. 1 with the estimated coefficients $\mathbf{c}^j$. It is important to note that, although $g_\theta(\cdot)$ initially takes joints predicted by $\Psi(\cdot)$ to reduce search space, $g_\theta(\cdot)$ learns to *refine* the noisy per-frame joint predictions into continuous joint flow using hints from the input RGB and skeleton graph structures (please also refer to Table. 3 for experimental results).

Once the joint flow $\{\Upsilon^j(\cdot)\}_{j=1}^J$ is obtained, we can propagate the learned joint flow to an arbitrary query point $\mathbf{p}$ via implicit LBS to compute the per-query pose flow $\Phi_\theta^{pose}(\cdot, \cdot) : \mathbb{R}^3 \times [0, 1) \to \mathbb{R}^3$:

$$\Phi_\theta^{pose}(\mathbf{p}, t) = \sum_{b=1}^B w_b(\mathbf{p}) \, \hat{\mathbf{T}}_{\theta, b}^t \, \mathbf{p}, \tag{3}$$

where $\hat{\mathbf{T}}_{\theta, b}^t$ is the rigid bone transformation matrix for $b$-th hand bone computed from the joint positions $\{\Upsilon^j(t)\}_{j=1}^J$ at time $t$ using a biomechanically valid conversion algorithm [16]. $w_b(\cdot) : \mathbb{R}^3 \to [0, 1]$ is a function that returns a pre-trained LBS weight for $b$-th hand bone given an input query point $\mathbf{p}$. Note that, as $\mathbf{p}$ is sampled from the canonical space (Sec. 3.1), our bone transformation matrices $\{\hat{\mathbf{T}}_{\theta, b}^t\}_{b=1}^B$ are computed *w.r.t.* to the canonical hand pose [16] to model the query correspondences between the canonical system and the posed system at time $t$.

**Shape flow.** Our shape flow models query-wise displacement flow *w.r.t.* the estimated pose flow. It aims to model spatio-temporal deformations (*e.g.*, soft tissue or identity-dependent deformations) that cannot be solely expressed by linear deformations *w.r.t.* hand articulation changes (Eq. (3)). Formally, our shape flow network $h_\omega(\cdot, \cdot) : \mathbb{R}^{T \times W \times H \times 3} \times \mathbb{R}^3 \to \mathbb{R}^{6N+3}$ learns a mapping from an input RGB sequence and a query point $\mathbf{p}$ to Fourier coefficients $\mathbf{c^P}$ representing the displacement flow of $\mathbf{p}$ as follows:

$$h_\omega\left((\mathbf{I}^t)_{t=1}^T, \mathbf{p}\right) = t_{\omega_2}(e_{\omega_1}((\mathbf{I}^t)_{t=1}^T, \mathbf{p})). \tag{4}$$

Similar to $g_\theta(\cdot)$, $h_\omega(\cdot, \cdot)$ is composed of two sub-networks[2] $e_{\omega_1}(\cdot, \cdot)$ and $t(\cdot)_{\omega_2}$ parameterized by the weights $\omega = \{\omega_1, \omega_2\}$. As shown in the lower branch of Fig. 2, our query encoder $e_{\omega_1}(\cdot, \cdot) : \mathbb{R}^{T \times W \times H \times 3} \times \mathbb{R}^3 \to \mathbb{R}^{T \times h'}$ extracts $h'$-dimensional query feature per time step. For each $t$, we create an initial query descriptor $\mathbf{f}^t \in \mathbb{R}^h$ by concatenating (1) a canonical query position $\mathbf{p}$, (2) a query position after applying the pose flow $\Phi_\theta^{pose}(\mathbf{p}, t)$ and its corresponding pixel-aligned feature [35], and (3) a skinning weight vector $w(\mathbf{p})$. This query descriptor is fed to an MLP-based query encoder to extract a query feature $\mathbf{f}'^t \in \mathbb{R}^{h'}$. Our temporal encoder $t_{\omega_2}(\cdot) : \mathbb{R}^{T \times h'} \to \mathbb{R}^{6N+3}$ then applies temporal convolutions to the per-time query features to predict the Fourier coefficients $\mathbf{c^P}$ representing the displacement flow of query $\mathbf{p}$. Finally, our shape flow $\Phi_\omega^{shape}(\cdot, \cdot) : \mathbb{R}^3 \times [0, 1) \to \mathbb{R}^3$ for a query point $\mathbf{p}$ is modeled by Eq. (1) using the query-wise Fourier coefficients $\mathbf{c^P}$ predicted by $h_\omega(\cdot, \cdot)$.

Our final flow $\Phi_{\theta, \omega}(\cdot, \cdot) : \mathbb{R}^3 \times [0, 1) \to \mathbb{R}^3$ for a query point $\mathbf{p}$ is obtained as a sum of the learned pose and shape flows, where both of them are represented by the Fourier series to enforce smooth query dynamics along the temporal axis:

$$\Phi_{\theta, \omega}(\mathbf{p}, t) = \Phi_\theta^{pose}(\mathbf{p}, t) + \Phi_\omega^{shape}(\mathbf{p}, t). \tag{5}$$

---

[1]Since the initial per-frame joints are discretized in time, their Fourier series coefficients are approximated by the numerical integration as discussed in [9]. We use *trapz* function in PyTorch [31] library.

[2]Although the input encoder and the temporal encoder in Eq. (2) and Eq. (4) have different network architectures, we use the same notations $e(\cdot)$ and $t(\cdot)$ for brevity.

**Loss functions.** Our network is trained using the (1) occupancy loss and (2) correspondence loss. First, the occupancy loss penalizes the deviation between the reconstructed and ground truth occupancy probabilities:

$$\mathcal{L}_{occ}(\theta, \omega) = \frac{1}{|\mathcal{X}_{occ}|} \sum_{\mathbf{p}, t, \mathcal{I} \in \mathcal{X}_{occ}} \|\hat{o}_{\theta, \omega}(\mathbf{p}, t, \mathcal{I}) - o_{gt}(\mathbf{p}, t, \mathcal{I})\|_2, \qquad (6)$$

where $\mathcal{X}_{occ}$ is a set of training examples and $\mathcal{I}$ is a sequence of RGB inputs. $\hat{o}_{\theta, \omega}(\mathbf{p}, t, \mathcal{I})$ and $o_{gt}(\mathbf{p}, t, \mathcal{I})$ denote the reconstructed and ground truth occupancy probabilities, respectively. Note that $\hat{o}_{\theta, \omega}(\mathbf{p}, t, \mathcal{I})$ is obtained by propagating the pre-trained canonical occupancy at $\mathbf{p}$ to the corresponding query position at $t$ using the query flow learned by our network. Since the canonical occupancy field is pre-trained and fixed, this occupancy loss enforces our flow network to learn query trajectories that better propagate the canonical occupancies $s.t.$ the occupancies propagated to $t$ better resemble the ground truth 3D shape at $t$.

Second, our correspondence loss penalizes the deviation between the reconstructed query position $\hat{p}_{\theta, \omega}(\mathbf{p}, t, \mathcal{I}) := \Phi_{\theta, \omega}(\mathbf{p}, t \,|\, \mathcal{I})$ and the ground truth query position $p_{gt}(\mathbf{p}, t, \mathcal{I})$ corresponding to $\mathbf{p}$ given a set of training examples $\mathcal{X}_{corr}$:

$$\mathcal{L}_{corr}(\theta, \omega) = \frac{1}{|\mathcal{X}_{corr}|} \sum_{\mathbf{p}, t, \mathcal{I} \in \mathcal{X}_{corr}} \|\hat{p}_{\theta, \omega}(\mathbf{p}, t, \mathcal{I}) - p_{gt}(\mathbf{p}, t, \mathcal{I})\|_2. \qquad (7)$$

Our final training objective is to minimize $\mathcal{L}(\theta, \omega) = \mathcal{L}_{occ}(\theta, \omega) + \lambda \mathcal{L}_{corr}(\theta, \omega)$, where $\lambda$ is a hyper-parameter to balance the influence between the two loss terms. For more details on network training and architecture, please refer to the supplementary section.

## 4 Experiments

**Datasets and evaluation metrics.** We use the two-hand (TH) and single-hand (SH) subsets of 30 FPS version InterHand2.6M [26] dataset, which contains diverse hand motions captured in RGB sequences with dense shape annotations. For each subset, we use samples annotated as *valid* hand type and follow the train/val/test splits of the original InterHand2.6M dataset. The resulting TH subset contains 477K training and 4K validation sequences, and SH subset contains 656K training and 5K validation sequences. To maintain consistency with the existing 4D continuous representations [29, 14, 15], we use sub-sequences of $T = 17$ frames sampled from the original sequences as inputs to our method. For testing, we use 2K sub-sequences (34K frames) randomly sampled from the test sequences of each subset. For qualitative evaluation, we additionally show our results on RGB2Hands [41] real dataset, which contains sequences of RGB hand motions without shape annotations. For evaluation metrics on 4D reconstruction, we use mean Intersection over Union (IoU) and Chamfer L1-Distance (CD) computed $w.r.t.$ the ground truth shapes of InterHand2.6M dataset. We also use L1 Correspondence Error (L1-Corr) for evaluating our learned shape correspondences.

**Compared methods.** As our method is the first RGB-based 4D continuous hand representation, there is no direct baseline. We thus compare ours to (1) RGB-based 3D hand reconstruction methods and (2) 4D implicit reconstruction methods modified to take RGB hands as inputs. For 3D hand reconstruction methods, we consider state-of-the-art methods on InterHand2.6M: ACR* [44], Two-Hand-Shape-Pose [45], IntagHand [19], and Im2Hands [18]. Note that ACR* [44] results are produced using the pre-trained ACR model officially released for in-the-wild demo. As these methods are proposed for two-hand reconstruction, we compare them on TH subset. For 4D representations, we consider (1) Occupancy Flow, which is the only 4D continuous representation that has shown results from RGB videos, and (2) LoRD, which is the most recent state-of-the-art 4D articulated implicit function proposed for human bodies. As LoRD originally takes SMPL [21] meshes fitted to 2.5D or 3D point cloud inputs, we have implemented a modified version (LoRD$^\dagger$) that uses MANO [34]-topology hand meshes reconstructed from RGB frames using IntagHand [19] to make direct comparisons. We compare these 4D implicit methods on both TH and SH subsets. For the off-the-shelf hand pose estimator $\Psi(\cdot)$ in our method, we adopt the joint estimation module of IntagHand [19] as in [18].

**Two-hand extension.** Our hand representation can be naturally extended to estimate 4D two-hand shapes to make comparisons on InterHand2.6M [26] TH subset. In a nutshell, we use two joint

flow networks each trained for left and right hands, respectively. The shape flow network is shared among both side of hands to implicitly capture correlation between them. Note that left and right side conditioning for shape flow estimation is done by taking as inputs a query position after applying the pose flow $w.r.t.$ the corresponding side of hand. Please find more details in the supplementary.

## 4.1 Video-Based 4D Hand Reconstruction

In Tab. 1, our method achieves state-of-the-art reconstruction results on InterHand2.6M [26]. The existing 3D reconstruction methods (3D Mesh and 3D Implicit categories) perform shape reconstruction for each frame independently. Thus, they often produce shapes with temporal jitters and do not reason about hand motions (*e.g.*, motion inter- and extrapolation). In 4D Implicit category, Occupancy Flow [29] achieves sub-optimal reconstruction quality due to the lack of shape articulation prior. While LoRD[†] [14] achieves better performance, it does not fully leverage pixel-aligned features for shape reconstruction. In Fig. 3, we also qualitatively show the reconstruction results, where ours produces the most plausible shapes that are also well-aligned to the input RGB observations.

Table 1: Quantitative comparisons of video-based 4D hand reconstruction on InterHand2.6M [26] dataset. Our method achieves state-of-the-art results on both TH and SH subsets.

| Subset | Category | Method | IoU (%) ↑ | CD (mm) ↓ |
|---|---|---|---|---|
| TH | 3D Mesh | ACR[*] [44] | 45.2 | 7.89 |
| | | Two-Hand-Shape-Pose [45] | 48.7 | 6.58 |
| | | IntagHand [19] | 57.5 | 5.27 |
| | 3D Implicit | Im2Hands [18] | 61.8 | 4.59 |
| | 4D Implicit | Occupancy Flow [29] | 32.4 | 18.27 |
| | | LoRD[†] [14] | 58.0 | 4.89 |
| | | FOURIERHANDFLOW (Ours) | **62.8** | **4.46** |
| SH | 4D Implicit | Occupancy Flow [29] | 44.7 | 13.21 |
| | | LoRD[†] [14] | 63.3 | 4.38 |
| | | FOURIERHANDFLOW (Ours) | **65.8** | **3.90** |

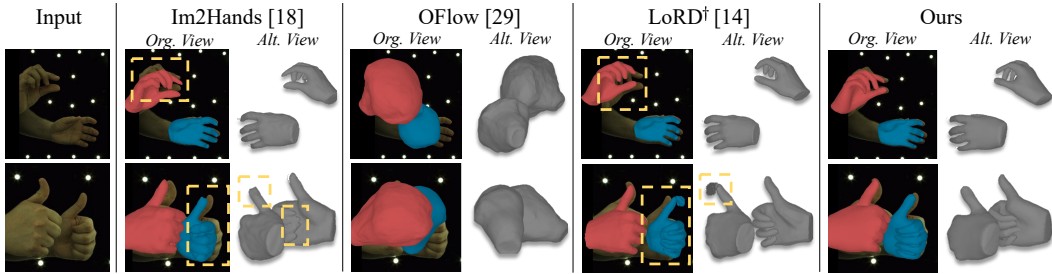

| Input | Im2Hands [18] | | OFlow [29] | | LoRD[†] [14] | | Ours | |
|---|---|---|---|---|---|---|---|---|
| | *Org. View* | *Alt. View* | *Org. View* | *Alt. View* | *Org. View* | *Alt. View* | *Org. View* | *Alt. View* |

Figure 3: Qualitative comparisons of video-based 4D hand reconstruction on InterHand2.6M [26] TH subset. **Please also refer to our supplementary for full-length video-based reconstruction comparisons, where ours achieves more favorable results (*e.g.* without temporal jitters).**

In Tab. 2 (columns 3-4), we also compare training and inference time between the implicit representations. Note that we use InterHand2.6M TH subset hereinafter unless specified. Im2Hands [18] is slow in both training and inference due to the global shape encoding and decoding step for shape refinement. While LoRD[†] [14] is the fastest in training, its inference is slow due to the test-time model optimization and per-time occupancy decoding. Occupancy Flow [29] achieves fast inference thanks to its simple network architecture, but its efficiency is still sub-optimal due to the use of an ODE solver [5]. Our method achieves the fastest inference time using the Fourier query flow representation. In Tab. 2 (column 5), we also compare the shape correspondence error between the implicit hand functions. Note that Im2Hands [18] and LoRD[†] [14] do not capture temporal shape correspondences due to per-time occupancy decoding. Compared to Occupancy Flow [29], we learn more accurate shape correspondences by articulation-aware query flow modeling.

Table 2: Training / inference time and learned correspondence comparisons among implicit representations. Training time is measured for each training iteration with a batch size of 1 and inference time is measured for each frame to make comparisons with the frame-based method [18]. Each time result is obtained as an average of 1K measurements.

| Category | Method | Training Time (sec.) ↓ | Inference Time (sec.) ↓ | L1-Corr (mm) ↓ |
|---|---|---|---|---|
| 3D Implicit | Im2Hands [18] | 3.31 | 6.62 | N/A |
| 4D Implicit | Occupancy Flow [29] | 6.15 | 0.26 | 16.8 |
| | LoRD† [14] | **0.15** | 114.72 | N/A |
| | FOURIERHANDFLOW (Ours) | 2.75 | **0.22** | **10.8** |

In addition, we compare our learned joint flow with the noisy per-frame joints estimated by $\Psi(\cdot)$ [19]. In Tab. 3, our joint flow network is shown to be effective in refining the initial per-frame joint predictions using hints from the input RGB frames and hand skeleton graph structures.

Table 3: Joint refinement results on Mean Per Joint Position Error (MPJPE).

| Method | MPJPE (mm) ↓ |
|---|---|
| Before Ref. | 13.3 |
| After Ref. | **11.0** |

## 4.2 Additional Experimental Results

**Texture transfer.** In Fig. 4, we show our results on texture transfer using the learned correspondences of implicit shapes. Given the reference texture defined in our canonical hand field (Sec. 3.1), we achieve high-quality texture transfer results on the hand reconstructions at randomly sampled sequences and time stamps. Note that, as our canonical hand field is shared among all sequences (*cf.* [29]), we can naturally obtain dense correspondences between inter-sequence hand reconstructions.

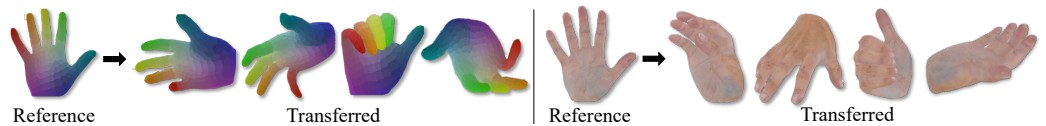

Reference    Transferred    Reference    Transferred

Figure 4: Texture transfer results using the learned correspondences between our hand reconstructions. Best viewed with 300% zoom-in.

**Motion inter- and extrapolation.** In Fig. 5, we show our motion inter- and extrapolation results. Given RGB frames observed at each time step, we sample hand shapes at inter- and extrapolated time values from the learned Fourier query flows. Our sampled shapes are shown to model smooth temporal evolution of hand shapes.

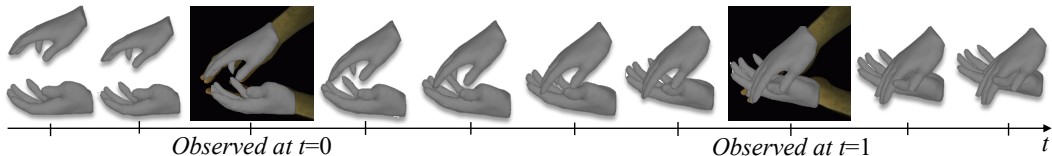

*Observed at t=0*        *Observed at t=1*        *t*

Figure 5: Our motion inter- and extrapolation results. Given RGB frames observed at each time step, our method can sample shapes at inter- and extrapolated time values.

**Generalization test.** In Fig. 6, we show additional qualitative results on RGB2Hands [41] real images using our model trained on Inter-Hand2.6M [26] dataset. Our model is shown to produce plausible reconstructions even from *unseen* RGB2Hands images, demonstrating its generalization ability.

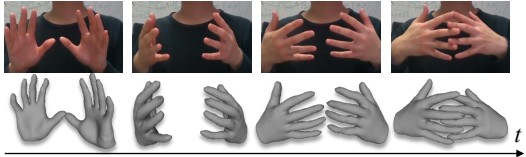

Figure 6: Generalization test on unseen RGB2Hands [41] real images.

### 4.3 Ablation Study

In Tab. 4, we compare video-based 4D hand reconstruction results among the variations of our method. *w/o Pose Flow* and *w/o Shape Flow* (rows 2-3) denote our method variations in which pose or shape flow is not modeled, respectively. It is shown that each type of flow contributes to more accurate reconstruction, thus the best accuracy is achieved when both flows are used. *Fourier → ODE* and *Fourier → InputCond* (rows 4-5) are variations in which (1) query flows are modeled by solving ODE [5] and (2) query positions are directly estimated conditioned on a time value, respectively. Our method based on Fourier query flow achieves the best performance, showing the effectiveness of flow estimation in the frequency domain (*cf.* spatial domain).

Table 4: Quantitative comparisons of video-based 4D hand reconstruction among the variations of our method on InterHand2.6M [26] TH subset. Our proposed method achieves the best results.

| Method | IoU (%) ↑ | CD (mm) ↓ | L1-Corr (mm) ↓ |
|---|---|---|---|
| w/o Pose Flow | 50.1 | 7.86 | 20.9 |
| w/o Shape Flow | 57.9 | 5.06 | 12.1 |
| Fourier → ODE | 60.4 | 4.93 | 11.5 |
| Fourier → InputCond | 61.0 | 4.87 | 11.0 |
| FOURIERHANDFLOW (Ours) | **62.8** | **4.46** | **10.8** |

## 5 Conclusion

We present FOURIERHANDFLOW, which is the first neural 4D continuous representation for human hands learned from RGB videos. We combine a continuous 3D hand occupancy field together with query flows learned as Fourier coefficients to model smooth and continuous temporal shape dynamics. To enable articulation-aware shape modeling, we introduce two types of Fourier query flow: (1) pose flow based on implicit linear blend skinning and (2) shape flow that models query-wise displacements. Our method achieves state-of-the-art results on video-based 4D hand reconstruction while being computationally more efficient than the existing implicit function-based methods.

**Broader impact.** Our method can be potentially used for video-based 4D reconstruction of articulated implicit shapes in general (*e.g.* human or animal bodies), which is useful for AR/VR applications.

**Limitations.** Although our method is more efficient than the state-of-the-art 3D and 4D implcit functions, its inference time is not yet sufficient to provide a real-time solution (*cf.* low-resolution mesh-based reconstruction methods [19, 44]). We plan to investigate ways to further improve our computational efficiency, *e.g.*, by adapting frequency-domain representation additionally along the spatial domain. Also, while Fourier query flow naturally models smooth temporal deformations, it constrains our query flow estimation on the low-mid frequency subspace. Thus, it may not model hand motions with actual temporal jitters, although they do not commonly occur. In addition, our method currently takes a fixed length of $T = 17$ sub-sequence as inputs following Occupancy Flow [29]. We will try to further address these limitations as our future work.

## Acknowledgments and Disclosure of Funding

This work was in part sponsored by NST grant (CRC 21011, MSIT), KOCCA grant (R2022020028, MCST) and IITP grant (2019-0-00075, MSIT). Minhyuk Sung acknowledges support from the NRF grant (RS2023-00209723) and IITP grants (2022-0-00594, RS-2023-00227592) funded by the Korean government (MSIT), the Technology Innovation Program (20016615) funded by the Korean government (MOTIE), as well as grants from ETRI, KT, NCSOFT, and Samsung Electronics.

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
