# FOURIERHANDFLOW: Neural 4D Hand Representation Using Fourier Query Flow — Appendix —

Jihyun Lee[1]    Junbong Jang[1]    Donghwan Kim[1]    Minhyuk Sung[1]    Tae-Kyun Kim[1,2]

[1]KAIST        [2]Imperial College London

{jyun.lee, junbongjang, kdoh2522, mhsung, kimtaekyun}@kaist.ac.kr

## S.1    Video Results

In https://youtu.be/gDnYcQni_Gk, we provide video results on (1) 4D reconstruction comparisons, (2) texture transfer, and (3) motion inter- and extrapolation. In 4D reconstruction comparisons, we compare our method with two baselines with the strongest quantitative results in Tab. 1 in the main paper: Im2Hands [11] and LoRD[†] [8]. We qualitatively show that our method reconstructs more accurate hand shapes with less jittery or abrupt temporal deformations. Please find more details in the video.

## S.2    Additional Ablation Study

**Low-pass filtering (LPF).** In our method, the joint flow network (Sec. 3.3 in the main paper) refines the initial Fourier coefficients of the per-frame joints predicted by an off-the-shelf pose estimator [12]. We additionally perform experiments using the initial Fourier coefficients for $N = 6$ basis functions *without the refinement*, which can be interpreted as applying simple low-pass filtering on the noisy joint trajectory. In Tab. S1 ($\text{Joint Flow} \rightarrow \text{LPF}$), we show that the reconstruction performance significantly decreases compared to that of our full method, which learns to effectively *refine* the initial joint trajectory in the frequency domain using hints from the input RGB frames and hand skeleton structures.

Table S1: Quantitative comparisons of video-based 4D hand reconstruction on InterHand2.6M [15] TH subset with low-pass filtering.

| Method | IoU (%) ↑ | CD (mm) ↓ | L1-Corr (mm) ↓ |
|---|---|---|---|
| Joint Flow → LPF | 56.5 | 5.84 | 13.9 |
| FOURIERHANDFLOW (Ours) | **62.8** | **4.46** | **10.8** |

**Number of basis functions ($N$).** We also investigate the effect of the number of basis functions, which is $N = 6$ in our main method. In Tab. S2, we show our reconstruction accuracy and efficiency using $N = \{4, 6, 8\}$ basis functions. Overall, our model performance is not affected much by $N$, but the performance is slightly decreased when $N$ is too small (*e.g.* $N = 4$). Thus, we choose $N = 6$ in our main method to achieve a good balance between accuracy and computational efficiency.

Table S2: Quantitative results of video-based 4D hand reconstruction on InterHand2.6M [15] TH subset with varying number of basis ($N$).

| $N$ | IoU (%) ↑ | CD (mm) ↓ | L1-Corr (mm) ↓ | Training Time (sec.) ↓ | Inference Time (sec.) ↓ |
|---|---|---|---|---|---|
| 4 | 62.5 | 4.50 | 10.9 | **2.50** | **0.12** |
| 6 (Ours) | **62.8** | 4.46 | **10.8** | 2.75 | 0.15 |
| 8 | 62.6 | **4.45** | **10.8** | 2.87 | 0.18 |

37th Conference on Neural Information Processing Systems (NeurIPS 2023).

These results imply that the function that we aim to reconstruct (i.e., the hand motion captured in InterHand2.6M [15]) is mostly smooth enough to be reconstructed with a number ($N = 6$) of Fourier terms, as our shape reconstruction accuracy does not *noticeably* increase when using a higher number of Fourier terms. However, if one has to reconstruct higher-frequency motions (other than the hand motions in InterHand2.6M), it would be desirable to use a higher number of Fourier terms ($N$) in our method.

**Off-the-shelf pose estimator ($\Psi$).** In the Tab. S3, we evaluate our 4D reconstruction results using two different models for $\Psi$ [12, 5]. While the initial pose accuracies have a considerably large gap (3.7mm in MPJPE), our final shape accuracies do not vary much, and both settings achieve the state-of-the-art 4D reconstruction results compared to the baseline methods in Tab. 1 in the paper. This demonstrates that our final shape accuracy is quite robust to the quality of the initial pose predictions.

Table S3: Quantitative results of video-based 4D hand reconstruction on InterHand2.6M [15] TH subset with varying the initial pose estimator ($\Psi$) architecture.

| Method | Initial Joint Error (MPJPE ↓) | Shape Accuracy (IoU ↑) | Shape Accuracy (CD ↓) |
|---|---|---|---|
| IntagHand [12] | 13.3 | 62.8 | 4.46 |
| DIGIT [5] | 17.0 | 62.2 | 4.58 |

**Error in initial pose estimation.** To additionally examine *what will happen if $\Psi$ provides bad predictions during testing*, we also measure our 4D reconstruction accuracy after injecting uniformly sampled noise with the amplitude $[-x, x]$ to every dimension of the initial pose prediction in the test time — following the exact ablation study setting in [9] (Section D in the supplementary). As shown in Table S4, our model performance is quite robust to the injected noises (gently decreasing its accuracy for a higher level of noise), and again, all settings achieve state-of-the-art results compared to the baselines in Tab. 1 in the paper.

Table S4: Quantitative comparisons of video-based 4D hand reconstruction on InterHand2.6M [15] with different degrees of noise ($x$) injection in initial pose prediction.

| Noise Level ($x$) | IoU (%) ↓ | CD (mm) ↓ |
|---|---|---|
| 0mm (No Noise) | 62.8 | 4.46 |
| 1mm | 62.7 | 4.48 |
| 3mm | 62.5 | 4.50 |
| 5mm | 62.2 | 4.53 |

## S.3   Additional Comparisons

**Comparisons with Fourier Occupancy Field [6].** Fourier Occpancy Field (FOF) [6] uses Fourier Series along one of the spatial axes (i.e., $z$-axis) to enable efficient 3D human reconstruction, while our method uses Fourier Series along the temporal axis (1) to regularize high-frequency (e.g., jittery) temporal shape change and (2) to enable efficient 4D hand reconstruction. Along with this difference, another important difference between FOF and our method lies in the characteristics of the shape category that each aims to reconstruct. FOF aims to reconstruct clothed humans, which have complex shape variations (e.g., cloth wrinkles) but with fewer self-occlusions caused by the underlying shape articulations. In contrast, our method aims to reconstruct human hands, which have more complex articulations leading to more severe self-occlusions. Thus, most of the existing *hand* implicit functions, e.g., [3, 9, 11], use articulated implicit representation to directly model pose-dependent deformations with respect to the learned canonical hand shape to incorporate strong pose prior. Similarly, one of our main contributions is also to propose articulation-aware query flows (i.e., pose and shape flows) to directly model pose-dependent 4D hand deformations. In Tab. S5, we show the experimental comparisons with FOF, where FOF yields similar results to our non-articulated implicit function baseline (Occupancy Flow [17]) in Tab. 1 in the paper.

**Comparisons with MANO [20] fitting.** We also compare ours to a classical MANO [20] fitting-based approach, where we first predict the keypoints and segmentation masks from the input RGB frames using [12] and then perform optimization-based MANO parameter fitting to those predictions. In Tab. S6, MANO fitting is shown to yield less accurate and efficient reconstruction results. This is because (1) the accuracy of the MANO fitting is directly bound to the accuracy of the intermediate predictions and (2) the inference involves test-time optimization.

Table S5: Quantitative comparisons of video-based 4D hand reconstruction on InterHand2.6M [15] TH subset with Fourier Occupancy Field (FOF) [6].

| Method | IoU (%) ↑ | CD (mm) ↓ |
|---|---|---|
| FOF [6] | 30.4 | 23.88 |
| FOURIERHANDFLOW (Ours) | **62.8** | **4.46** |

Table S6: Quantitative comparisons of video-based 4D hand reconstruction on InterHand2.6M [15] TH subset with a MANO [20] fitting-based approach.

| Method | IoU (%) ↑ | CD (mm) ↓ | Inference Time (sec.) ↓ |
|---|---|---|---|
| MANO [20] Fitting | 43.7 | 6.48 | 28.4 |
| FOURIERHANDFLOW (Ours) | **62.8** | **4.46** | **0.22** |

**Comparisons on joint estimation.** Although the primary goal of this research is accurate shape reconstruction rather than joint estimation, we additionally evaluate our results on MPJPE in comparison to [12, 26, 27] in Tab. S7. As [12, 27] originally use the additional ground truth bone length for rescaling the joint predictions, we use the mean bone length of the hands in the training set of InterHand2.6M [15] following [11] to perform fair comparisons. Our method achieves lower MPJPE than the compared methods on InterHand2.6M dataset.

Table S7: Quantitative comparisons of joint estimation on InterHand2.6M [15] TH subset.

| Method | ACR* [26] | Two-Hand-Shape-Pose [27] | IntagHand [12] | Ours |
|---|---|---|---|---|
| MPJPE (mm) ↓ | 18.3 | 15.9 | 13.3 | **11.0** |

## S.4 Details on the Pre-Trained Occupancy and Implicit Linear Blend Skinning Weight Fields

### S.4.1 Background: Linear Blend Skinning (LBS)

We first review linear blend skinning (LBS), which is a widely-used technique to deform a shape with underlying skeletal structures. It is originally used for deforming a *mesh* according to rigid bone transformations [13, 20]. Given a mesh with a set of $V$ initial vertex positions $\{\hat{\mathbf{v}}^{(i)}\}_{i=1}^{V}$ and rigid transformation matrices of $B$ bones $\{\mathbf{T}_b\}_{b=1}^{B}$, the deformed vertex positions $\{\mathbf{v}^{(i)}\}_{i=1}^{V}$ are computed as:

$$\mathbf{v}^{(i)} = \sum_{b=1}^{B} \mathbf{w}_b^{(i)} \mathbf{T}_b \hat{\mathbf{v}}^{(i)}, \quad \forall i = 1, ..., V, \tag{1}$$

where $\mathbf{w}^{(i)} \in [0, 1]^B$ is a skinning weight vector for $\hat{\mathbf{v}}^{(i)}$ *s.t.* $\sum_{b=1}^{B} \mathbf{w}_b^{(i)} = 1$. Each entry $\mathbf{w}_b^{(i)}$ represents the amount of influence that $\mathbf{T}_b$ has on the deformed position of $i$-th vertex.

Recently, neural implicit 3D shape representations [14, 22] have also adopted LBS for articulated shape modeling. Since LBS weights should be implicitly defined given an arbitrary 3D query position, they use a neural network $w(\cdot) : \mathbb{R}^3 \to [0, 1]^B$ to learn the mapping from an input query $\mathbf{p} \in \mathbb{R}^3$ to the corresponding skinning weight vector $\mathbf{w^p} \in [0, 1]^B$. Analogous to Eq. (1), the deformed query position is computed by applying the weighted average of the transformations $\{\mathbf{T}_b\}_{b=1}^{B}$, where the weights are determined by $\mathbf{w^p}$.

### S.4.2 Learning Canonical Hand Occupancy and Implicit LBS Weights

We now provide details on pre-training the hand occupancy and LBS weight fields used in our method (Sec. 3.1 in the main paper), for which we utilize a modified version of LEAP [14] model. LEAP is originally proposed for learning the occupancy of human bodies from a set of bone transformation inputs $\{\mathbf{T}_b\}_{b=1}^{B}$. It first learns forward and inverse LBS functions using neural networks. Then, it uses (1) the cycle-distance feature computed via forward and inverse LBS and (2) the point feature computed using the three types of encoders (*i.e.* shape, structure, and pose encoders) to predict the occupancy at the query point (please refer to [14] for more details).

Note that our method only requires a forward LBS function $w(\cdot) : \mathbb{R}^3 \to [0, 1]^B$, which maps a query point in the *canonical* space to a skinning weight vector, and the occupancy function $o(\cdot) : \mathbb{R}^3 \to [0, 1]$ that models the canonical hand shape. For the forward LBS function $w(\cdot)$, we directly adopt the architecture of the forward LBS network of LEAP. While the original network takes the canonical SMPL [13] vertices to extract a shape feature for the canonical *human body* shape, our network takes the canonical MANO [20] vertices as inputs to encode the canonical *hand* shape. Note that our method is not dependent on MANO model except for this single set of canonical MANO hand vertices. For the occupancy function $o(\cdot)$, we have empirically found that using the point feature extracted from the structure encoder is sufficient to obtain a decent-quality canonical hand occupancy field. Thus, we have removed the shape and pose encoders and the cycle-distance feature from the occupancy network of LEAP for computational efficiency. For the structure encoder, we use the kinematic structure of hands instead of human bodies. Other architecture or training details are unchanged from the original LEAP network.

## S.5 Reproducibility

We now report the implementation details for the reproducibility of our method. Note that minor implementation details will be also available through our code, which will be published after the review period.

### S.5.1 Network Architecture

**Joint flow network.** For inputs to our joint encoder $e_{\theta_1}(\cdot)$, we build initial hand-skeleton graphs $\{\mathcal{G}^t\}_{t=1}^T$, where $T = 17$. For each $\mathcal{G}^t$, we use $J = 21$ nodes each corresponding to a hand joint at time $t$ with edges of a hand skeleton structure (see the upper branch of Fig. 2 in the main paper). Each node feature is initially set as the concatenation of the 3D position and pixel-aligned feature [21] of the corresponding hand joint estimated by the off-the-shelf pose estimator [12]. For the image encoder used to extract the pixel-aligned features, we adopt the stacked hourglass architecture [16] with batch normalization [7] replaced with group normalization [25]. We use two stacks with a feature channel size of 256, except for the output feature channel size set as 128. The 3D joint position is also augmented using a positional encoder, which is a single linear layer (which is shared for all joints) that outputs a 128-dimensional positional feature. The concatenated node features are in the form of $\mathbb{R}^{T \times J \times (128+128+3)}$.

For the joint encoder $e_{\theta_1}(\cdot)$, we use a network composed of three graph convolution blocks with inter-block residual connections. Each block consists of two Chebyshev spectral graph convolution layers [4] with a Chebyshev order of 2 and a feature channel size of 128. Each layer is followed by layer normalization [2] and ReLU [1] activation. The resulting node features are in the form of $\mathbb{R}^{T \times J \times 128}$.

Our temporal encoder $t_{\theta_2}(\cdot)$ then applies a shared convolutional neural network for each joint along the temporal dimension to extract per-joint temporal features. The layer configurations for each temporal convolutional layer can be found in Tab. S8 (rows 2-4). The output per-joint temporal feature is concatenated with the Fourier coefficients of the corresponding joint predicted by an off-the-shelf pose estimator [12] and fed to a multilayer perceptron (MLP) for refined coefficients prediction. The layer configurations for each fully-connected layer can be also found in Tab. S8 (rows 5-7).

Table S8: Layer configurations of the temporal encoder in our joint flow network. Each layer (except the last *MLP-3* layer) is followed by ReLU [1] activation.

|        | Layer Description | Output Dimension |
|--------|-------------------|------------------|
| TCNN-1 | (Temporal conv., filter size 3, 96 features, stride 2) | $J \times 96 \times 8$ |
| TCNN-2 | (Temporal conv., filter size 3, 64 features, stride 2) | $J \times 64 \times 3$ |
| TCNN-3 | (Temporal conv., filter size 3, 64 features, stride 1) | $J \times 64 \times 1$ |
| MLP-1  | (Linear, 128 features) | $J \times 128$ |
| MLP-2  | (Linear, 64 features) | $J \times 64$ |
| MLP-3  | (Linear, 14 $(= 6N + 3)$ features) | $J \times 14 (= 6N + 3)$ |

**Shape flow network.** For inputs to our query encoder $e_{\omega_1}(\cdot)$, we create query descriptors $\{\mathbf{f}^t\}_{t=1}^T$, where $T = 17$. For each $\mathbf{f}^t$, we concatenate (1) the query position $\mathbf{p}$ and (2) the query position

after applying the previously estimated pose flow $\Phi_\theta^{pose}(\mathbf{p}, t)$ and its pixel-aligned feature [21], and a skinning weight vector of $\mathbf{p}$ predicted by the pre-trained LBS weight function $w(\cdot)$. For the image encoder used to extract the pixel-aligned features, we adopt the same network architecture as the image encoder in the joint flow network. The resulting query descriptor is in the form of $\mathbb{R}^{3+3+128+16}$. For the query encoder $e_{\omega_1}(\cdot)$, we use a network composed of two fully-connected layers with a feature channel size of 128. Each layer is followed by ReLU [1] activation. For the temporal encoder $t_{\omega_2}(\cdot)$ that predicts the Fourier coefficients of the query-wise displacement flow, we use a convolutional neural network applied along the temporal dimension, whose layer configurations can be found in Tab. S9.

Table S9: Layer configurations of the temporal encoder in our shape flow network. Each layer (except the last *TCNN-3* layer) is followed by weight normalization [23] and ReLU [1] activation. $B$ denotes the query batch size.

| | Layer Description | Output Dimension |
|---|---|---|
| TCNN-1 | (Temporal conv., filter size 3, 128 features, stride 2) | $B \times 128 \times 8$ |
| TCNN-2 | (Temporal conv., filter size 3, 128 features, stride 2) | $B \times 128 \times 3$ |
| TCNN-3 | (Temporal conv., filter size 3, 14 ($= 6N + 3$) features, stride 1) | $B \times 14(= 6N + 3) \times 1$ |

**Two-hand extension.** We now explain the architecture of the two-hand version of our method, which is used for the experiments on the two-hand (TH) subset of InterHand2.6M [15] dataset and RGB2Hands [24] dataset. For the joint flow network, we use two networks each trained for left and right hands, respectively. For the shape flow network, we use one shared network to implicitly capture the correlation between left and right hands through a shared feature embedding space. Left and right conditioning is incorporated when creating an initial query descriptor $\mathbf{f}^t$ in two ways. First, we use the query position after applying the pose flow *w.r.t.* the corresponding side of the hand and its pixel-aligned feature [21]. Second, we concatenate a binary label – [1, 0] for left side and [0, 1] for right side – to the query descriptor. Other implementation details are unchanged from the single-hand version of our method.

**Off-the-shelf pose estimator [12].** For the off-the-shelf hand pose estimator $\Psi(\cdot)$, we use the joint estimation module of IntagHand [12] similar to Im2Hands [11]. As IntagHand estimates two-hand joints, we use the original IntagHand network for the experiments on the two-hand (TH) subset of InterHand2.6M [15] dataset and RGB2Hands [24] dataset. For the experiments on the single-hand (SH) subset of InterHand2.6M dataset, we remove the cross hand attention module to perform single-hand joint estimation. Other architecture or training details are unchanged from the original IntagHand network. Similar to Im2Hands [11], we note that our method is agnostic to the architecture of the off-the-shelf pose estimator, thus it is possible to use any other pose estimator.

### S.5.2 Training Details and Datasets

**Training details.** Note that our method first predicts the joint flow and then predicts the shape flow dependent on the estimated joint flow. Thus, to enable more robust training, we first (1) train the joint flow network and then (2) train the shape flow network while freezing the parameters of the joint flow network. We train both networks for $100K$ training steps using an Adam [10] optimizer with a learning rate of $1e-4$. As the joint flow network itself does not perform dense shape estimation, we only use correspondence loss $\mathcal{L}_{corr}(\cdot, \cdot)$ *w.r.t.* the ground truth hand joint positions when training the joint flow network. For training the shape flow network, we use both the correspondence loss $\mathcal{L}_{corr}(\cdot, \cdot)$ and the occupancy loss $\mathcal{L}_{occ}(\cdot, \cdot)$ with the value of hyper-parameter $\lambda$ set as 10. Training on a single RTX 4090 GPU takes about 1 day and 3 days for the joint flow network and the shape flow network, respectively.

**Datasets.** For InterHand2.6M [15] dataset, we follow the data pre-processing steps used in Intag-Hand [12]. For selecting test subsequences of length $T = 17$, we randomly choose 2K starting frames from each subset – with a random seed fixed for all experiments – and additionally collect the following 16 frames. We plan to release the specific test data configurations along with the code.

**Data normalization.** Along the spatial dimension, we normalize the 3D coordinate space of each input sequence by aligning the predicted hand root joint of the first frame to the origin point. Along the temporal dimension, we do not apply normalization.

### S.5.3 Modification of LoRD [8]

As also briefly mentioned in Sec. 4 in the main paper, the original LoRD [8] network learns 4D humans conditioned on the input 2.5D or 3D point cloud sequence and SMPL [13] meshes fitted to the inputs. Since the goal of our work is to learn 4D continuous representation from RGB frame sequences, we condition the prediction of LoRD on the MANO [20]-topology hand meshes predicted from the input RGB sequence to make direct comparisons. We use IntagHand [12] for the mesh prediction, since it has shown the strongest quantitative results among the image-based reconstruction methods that output fixed-topology meshes (see *3D Mesh* category in Tab. 1 in the main paper). We use the predicted hand meshes for both local part tracking and test-time optimization of LoRD. We also note that the other 4D continuous representations use network architectures (*e.g.* [18, 19]) that specifically takes point cloud inputs, thus it is non-trivial to adapt them to learn directly from RGB sequence inputs. To the best of our knowledge, ours is the first *articulation-aware* 4D continuous representation proposed for RGB sequence inputs.