# OpenReview forum: "FourierHandFlow: Neural 4D Hand Representation Using Fourier Query Flow"
_NeurIPS.cc/2023/Conference — NeurIPS 2023 poster_

### Official Review · Reviewer_vgcD · 2023-07-05

**Soundness:** 3 good
**Presentation:** 4 excellent
**Contribution:** 3 good
**Rating:** 6
**Confidence:** 5

**Summary:**

The paper proposes a method to reconstruct 4D hand (3D hand sequence) from a short RGB sequence with two types of Fourier Query Flow (pose flow and shape flow). In the Fourier Query Flow, the 3D trajectory of each point is transformed into 3 Fourier series along the time dimension and represented with the first few coefficients (3*(2*6+1) = 39 in this paper). Pose flow is generated with joint flow via LBS. The geometry is represented in the canonical space with a pretrained occupancy network and warped into the real space with the pose flow. Then shape flow adds small displacements to it. Experiments demonstrate that the proposed method outperforms existing methods and can produce continuous and smooth results.

**Strengths:**

Originality: The idea of represent 3D flow with Fourier series is novel and interesting.

Quality: The presented results is of high-quality.

Clarity: The paper is well-written and easy to follow.

Significance: It seems that this method prevents the results from temporal jitters or abrupt motions, which I’ll discuss in Questions. And it’s very computational efficient.


**Weaknesses:**

I do not see some major weakness. But I still have some questions about this method. I’ll leave them to the Question part. And it would be great if authors can add these discussions to the paper.

A typo: line 228 t()^2 ?


**Questions:**

1.	It seems that using the low frequency representation can prevent temporal jitters or abrupt motions. However, lack of high frequency parts may lead to Gibbs phenomenon. So, I want to see some discussion about this. Does your method still work in the swift movement situation?
2.	It seems that the shape flow is working in the real space (not the canonical space), which is different from some methods, like MANO. So, I wonder if it works for different hand shapes, like longer finger or wider palm. If it works, I’d like to see some examples with different hand shapes.
3.	How about applying Fourier Query Flow to MANO model directly but not a pretrained occupancy network? I don’t see why using occupancy network would be better in this scenario. I’d like to see more discussion on this.


**Limitations:**

I appreciate the authors’ discussions about limitations of the proposed approach in Sec. 5, which helps the understanding of the suitable scenarios.

---

> ### Author Rebuttal · Authors · 2023-08-06
>
> We thank the reviewers for their thoughtful feedback and finding that the proposed method is “novel and interesting” [vgcD] and shows “large improvements” [rxcm, g8GH]. We also appreciate [rxcm] for finding that the “paper is clearly written and well-motivated.” In what follows, we address the concerns of the reviewers.
>
> ---
>
> > **Weakness 1.** "A typo: line 228 t()^2 ?"
>
> **Reply:** ^2 denotes *footnote 2*. We will change the footnote symbol for better clarity in the revision. Thank you for pointing this out.
>
> > **Question 1.** “It seems that using the low frequency representation can prevent temporal jitters or abrupt motions. However, lack of high frequency parts may lead to Gibbs phenomenon. So, I want to see some discussion about this. Does your method still work in the swift movement situation?”
>
> **Reply:** While Gibbs phenomenon *theoretically* occurs when approximating a discontinuous function using a finite Fourier Series, we have observed that the function that we aim to reconstruct -- the hand motion captured in InterHand2.6M [26] -- is mostly smooth enough to be reconstructed with a number ($N=6$) of Fourier terms. In Table S1 in the supplementary, we quantitatively showed that our shape reconstruction accuracy does not noticeably increase when using a higher number of Fourier terms than $6$ on InterHand2.6M dataset. Also, *Sequences 1 and 2* of our supplementary video (0:13-0:42) qualitatively show one of the swiftest hand motions captured in the dataset, and we have not *noticeably* observed the Gibbs phenomenon in the reconstructed motion. However, if one has to reconstruct higher-frequency motions (other than the hand motions in InterHand2.6M), it would be desirable to use a higher number of Fourier terms ($N>6$) in our method. We will add this discussion in the revision.
>
> > **Question 2.** “It seems that the shape flow is working in the real space (not the canonical space), which is different from some methods, like MANO. So, I wonder if it works for different hand shapes, like longer finger or wider palm. If it works, I’d like to see some examples with different hand shapes.”
>
> **Reply:** As you mentioned, our hand shape reconstruction itself is not constrained by MANO [34] PCA subspace. However, we have found that the hands captured in InterHand2.6M [26] dataset do not contain significant shape variations across the identities. Although our implicit function-based method leads to better reconstruction results than the MANO-based prediction method [45] in Table 1 in the paper, we will try to show reconstruction examples with more diverse hand shapes using the *very recently-released* dataset [R1] containing more hand shape variations. Thank you for your valuable comment.
>
> [R1] Potamias *et al.*, Handy: Towards a high fidelity 3D hand shape and appearance model, In CVPR, 2023.
>
> > **Question 3.** “How about applying Fourier Query Flow to MANO model directly but not a pretrained occupancy network? I don’t see why using occupancy network would be better in this scenario. I’d like to see more discussion on this.”
> >
>
> **Reply:** We thought that one of the important advantages of Fourier Query Flow is its ability to inherently model the **continuous** shape deformations along the temporal axis, e.g., allowing temporal shape inter- and extrapolation (Figure 5 in the paper). Although it is still possible to apply Fourier Query Flow to the MANO model (which is spatially discretized) directly, we wanted to preserve the continuity in the representation along both the spatial and the temporal axes — to keep the consistency between the axes and also to follow the motivation of the existing 4D continuous representations [14, 15, 29]. For example, we have observed that our method can generate more plausible hand shapes due to the capability for reconstruction in an arbitrary spatial resolution -- as shown in Figure R1 (in PDF). We will make sure to add this discussion in the revision.

---

> > ### Comment · Reviewer_vgcD · 2023-08-18
> > **Comment to the rebuttal**
> >
> > I thank authors' time and effort to answer my questions.
> >
> > The authors' replies address most of my concerns. However, there is still one thing I want to point out.
> >
> > In my Question 2, what I really concern is why p = LBS(p') + \delta p (this is a simple version of Equation 5 in your paper)，but not p = LBS(p' + \delta p). Why is it designed this way? Theoretically, the former one won't work well on shape variations. And I think Fourier-->inputCond in your experiments is not the latter one ( p = LBS(p' + \delta p) ), which is what I mean.
> >
> > So, I maintain my score for this work.

---

> > > ### Author Response · Authors · 2023-08-18
> > >
> > > Thank you for your valuable comment. Let Eq. (1) be our current shape formulation $\mathbf{p} = \mathrm{LBS}(\mathbf{p}’) + \Delta \mathbf{p}$, and let Eq. (2) be the suggested shape formulation $\mathbf{p} = \mathrm{LBS}(\mathbf{p}’ + \Delta \mathbf{p})$.
> > >
> > > The main difference between Eq. (1) and Eq. (2) lies in whether to predict the shape variations (modeled by $\Delta \textbf{p}$) in the unposed space (before applying LBS) or in the posed space (after applying LBS). Since our method performs image-based hand reconstruction using pixel-aligned features, we thought that modeling such shape variations in the posed space — which is more directly aligned with the input image space — would be more effective. Also, the shape formulation in Eq. (2) would require our LBS network to learn implicit LBS fields dependent on the canonical shape variations (rather than one pre-trained LBS field as in our current method). Thus, we thought that the formulation in Eq. (1) would be a simpler solution.
> > >
> > > Yet, we will also try to investigate the suggested direction in our future work. Thank you again for your thoughtful comment.

---

### Official Review · Reviewer_s3gm · 2023-07-06

**Soundness:** 2 fair
**Presentation:** 1 poor
**Contribution:** 2 fair
**Rating:** 3
**Confidence:** 3

**Summary:**

This paper introduces FourierHandFlow - an implicit 4d representation for learning spatio-temporal hand shape deformations.
The core idea is to introduce a coarse-to-fine implicit deformation model parameterized with a fixed Fourier basis to ensure smoothness and efficient inference. The coarse (pose / joint flow) part models the dynamics of the joints conditioned on images and pose predictions, and the fine part models the dynamics of the per-query point deformations on top of the joint flow, conditioned on image features.
Experimental evaluation is conducted on InterHands2.6M and indicates that the proposed method outperforms recent baselines in terms of quality of shape reconstruction.


**Strengths:**

- Overall coarse-to-fine flow formulation generally makes sense, as most of the local per-point deformations would be strongly dependent on the joints.
- Proposed formulation uses a fixed Fourier basis parameterisation of the flows is interesting, and bears multiple benefits:
  a) it allows for more efficient inference, since re-sampling points over temporal domain can be computed in closed form for different timesteps.
  b) it is guaranteed to lead to smooth trajectories by construction.
- Implicit 4D formulation leads to automatic learning of correspondences, which allows texture transfer.
- Quantitative results indicate that the proposed method outperforms multiple recent baselines (although there is not enough clarity on the evaluation protocol, see below).

**Weaknesses:**

1. The paper is at times hard to follow.
- For example, when introducing the method (L136), authors do not really specify an exact form of the underlying representation, and vaguely refer to it taking a sequence of RGB frames as input. Does the overall method take any other sources of supervision or conditioning?
- Similarly, from Section 3.1 it is not really possible to understand what is the ground truth used to pre-train the occupancy and LBS functions. Do you have geometry ground truth here? What is the resolution of this ground truth (a short look on InterHand2.6 suggests that geometry that is provided is coming from MANO)?
- After going through the experimental section I am still not sure I understand what is the ground truth geometry that this paper is comparing against.

2. Motivation / quality.
- The quality of the resulting geometry seems to be of low resolution and does not contain any high-frequency details.
- The main argument for using more complex machinery of implicit 4D representations over explicit mesh-based representations (L108) is the ability of those to capture high-resolution geometry details.
- Yet, the resulting meshes seem to contain no high-resolution details, and examples e.g. in Figure 3 show that all the considered methods struggle at rough alignment with the images - which should be possible to solve with more robust keypoint / segmentation constraints.
- The question thus arises if there is enough motivation for the development of the complex implicit machinery. If the resolution is the only limiting factor, why not upsample the mesh of MANO and fit it to ground truth scans, followed by a simple image2shape regression?

3. Evaluation / baselines.
- There is no comparison to a parametric model MANO fitted to sparse constraints such as keypoints / segmentation.

**Questions:**

- Could you please clarify what exactly is the geometry you are using for training the occupancy/lbs models and for evaluation? If it is MANO fitted shapes, then, if your model does not have any appearance, I am confused how can it learn any more detailed reconstruction than the underlying low-resolution parametric model?
- Why is there no comparison to a simple MANO model fitted to whatever InterHand2.6m constraints are available?


**Limitations:**

Authors discuss limitations and broader societal impact.

---

> ### Author Rebuttal · Authors · 2023-08-06
>
> We thank the reviewers for their thoughtful feedback and finding that the proposed method is “novel and interesting” [vgcD] and shows “large improvements” [rxcm, g8GH]. We also appreciate [rxcm] for finding that the “paper is clearly written and well-motivated.” In what follows, we address the concerns of the reviewers.
>
> ---
>
> > **Weakness 1-1.** “For example, when introducing the method (L136), authors do not really specify an exact form of the underlying representation, and vaguely refer to it taking a sequence of RGB frames as input. Does the overall method take any other sources of supervision or conditioning?”
> >
>
> **Reply:** During training, our method takes as input a sequence of monocular RGB frames (lines 136-137) and outputs 4D hand shapes in the form of an occupancy field, which is supervised by the ground truth 4D hand geometry (i.e., a temporal sequence of 3D hand geometries) of InterHand2.6M [26] dataset (lines 258-260). During testing, we only use a sequence of RGB frames as input to our method, as our main goal is to reconstruct 4D human hands from monocular RGB sequences (lines 19-20, 36-37).
>
> > **Weakness 1-2.** “Similarly, from Section 3.1 it is not really possible to understand what is the ground truth used to pre-train the occupancy and LBS functions. Do you have geometry ground truth here? What is the resolution of this ground truth?”
> >
>
> **Reply:** The same InterHand2.6M [26] hand geometries are used for pre-training the occupancy and LBS functions. These ground truth shapes are given as meshes with the MANO [34] topology.
>
> > **Weakness 1-3.** “After going through the experimental section I am still not sure I understand what is the ground truth geometry that this paper is comparing against.”
> >
>
> **Reply:** We used "the ground truth shapes of InterHand2.6M dataset" (line 268-269).
>
> > **Weaknesses 2-1 - 2.3.** “[...] The main argument for using more complex machinery of implicit 4D representations over explicit mesh-based representations (L108) is the ability of those to capture high-resolution geometry details. Yet, the resulting meshes seem to contain no high-resolution details. [...]
> >
>
> > **Weakness 2-4.** “The question thus arises if there is enough motivation for the development of the complex implicit machinery. [...]”
> >
>
> > **Question 1.** "I am confused how can it learn any more detailed reconstruction than the underlying low-resolution parametric model?"
>
> **Reply:** In Figure 3 in the paper, we showed the qualitative results of ours in comparison to the same implicit function-based methods only. In Figure R1 (in PDF), we additionally show the reconstruction examples of the current state-of-the-art mesh-based method [19] and ours on InterHand2.6M [26] dataset. As implicit representations learn continuous shape fields, they can naturally reconstruct shapes in an arbitrary resolution (including the extrapolated resolution beyond the observed ground truth resolution). Following the existing methods [15, 16, 18, 24, 25, 29, 39] that similarly use the ground truth MANO [34] of SMPL [21] meshes for training implicit functions, we believe that using implicit representation introduces several advantages including (1) the aforementioned arbitrary-resolution resolution reconstruction and (2) more accurate shape modeling due to the utilization of the pixel-aligned features directly corresponding to **dense** query points. In Table 1 in the paper, we also experimentally demonstrated that our method reconstructs more accurate shapes than the existing methods based on MANO parameter [44, 45] or MANO vertex regression [19] on InterHand2.6M.
>
> About (1) using "keypoint / segmentation constraints" and (2) "upsampl[ing] the mesh of MANO and fit[ting] it to ground truth scans" mentioned in the review, please refer to our answer to the next questions.
>
> > **Weakness 3-1.** “There is no comparison to a parametric model MANO fitted to sparse constraints such as keypoints / segmentation.”
> >
>
> > **Question 2.** *“Why is there no comparison to a simple MANO model fitted to whatever InterHand2.6m constraints are available?”*
> >
>
> **Reply:** We would like to kindly remind you that **we aimed to perform 4D hand reconstruction from monocular RGB sequences in this paper**. Thus, **only RGB observations are available as inputs to our method — making it non-trivial to directly perform MANO fitting to "keypoints", "segmentation", or "ground truth scans" as mentioned**. Thus, for experimental comparisons, we mainly considered the existing methods [18, 19, 44, 45] with the state-of-the-art *RGB-based* reconstruction results on InterHand2.6M [26]. Although it is possible to consider taking a multi-stage-based approach, in which we first predict the keypoints and segmentation masks from the input RGB frames and then perform optimization-based MANO fitting to those predictions, the accuracy of the fitted MANO would be directly bounded to the accuracy of the intermediate predictions. Also, considering that one of the objectives of this paper was the *"computational efficiency of 4D reconstruction"* (lines 41-42), a MANO fitting-based approach would lead to a less efficient solution due to the need for test-time optimization. In the table below, we show the experimental results of such MANO fitting using the state-of-the-art keypoint and segmentation predictor (IntagHand [19]) on InterHand2.6M [26] dataset, which yields less accurate and less efficient reconstruction results than ours. For the comparisons with the existing methods based on the feedforward MANO parameter prediction [45] or MANO vertex coordinates prediction [19] directly from images - which also intermediately utilize the *predicted* joints and segmentation priors - please refer to Table 1 in the paper.
>
> |  | Shape accuracy (IoU $\uparrow$) | Shape accuracy (CD $\downarrow$) | Inference time (sec. $\downarrow$) |
> | --- | --- | --- | --- |
> | **MANO [34] fitting** | 43.7  | 6.48 | 28.4 |
> | **Ours** | 62.8 | 4.46 | 0.22 |

---

> > ### Comment · Reviewer_s3gm · 2023-08-15
> >
> > Thank you for the detailed response! I am not sure it is clear to me how the method could learn anything beyond the ground truth supervision (which are MANO shapes). The fact that you are using richer conditioning signals could help only in the case of either having appearance or scans (multi-view reconstructions) _as supervision_ ? Otherwise, the upper bound for what the model can learn will always be MANO model (since it is the model that generated the ground truth?).

---

> > > ### Author Response · Authors · 2023-08-18
> > >
> > > As you mentioned, the shape details learned by our method may be bounded to the ground truth shapes (which are MANO shapes in the case of InterHand2.6M [26]) used in model training. However, we again clarify that our main motivation for using implicit representation is its ability (1) to learn accurate pixel-aligned shapes and (2) to model shapes in an arbitrary resolution. For (1) learning accurate pixel-aligned shapes, we showed (in Table 1 in the paper and our rebuttal experiment) that our implicit representation-based method leads to more accurate shape-fitting results with respect to the ground truth MANO meshes of InterHand2.6M [26] — compared to the direct MANO parameter or MANO vertex regression methods [19, 44, 45]. This is because we utilized “the pixel-aligned features directly corresponding to dense query points” as mentioned in our rebuttal. For (2) modeling shapes in an arbitrary resolution, we followed the motivation of the existing methods [15, 16, 18, 24, 25, 29, 39] that similarly use the ground truth MANO [34] of SMPL [21] meshes for training implicit functions — which discussed its ability to adaptively control the shape resolution in test time based on the application needs (rather than one fixed resolution as in the existing mesh-based reconstruction methods [19, 44, 45]). We think that particularly learning high-frequency shape details (e.g., hand wrinkles) is a different matter, and *we did not originally claim this anywhere in the paper* — though it is achievable given more detailed GT shapes (similar to reconstructing clothing details in human bodies using implicit representation as in, e.g., [R1, R2, R3]).
> > >
> > > [R1] Saito *et al.*, PIFu: Pixel-Aligned Implicit Function for High-Resolution Clothed Human Digitization, ICCV 2019
> > >
> > > [R2] Xiu *et al.*, ICON: Implicit Clothed Humans Obtained from Normals, CVPR 2022
> > >
> > > [R3] He *et al.*, Geo-PIFu: Geometry and Pixel Aligned Implicit Functions for Single-view Human Reconstruction, NeurIPS 2022

---

> > > > ### Comment · Reviewer_s3gm · 2023-08-18
> > > >
> > > > Thanks for the response. It sort of confirms my concern that the method is bounded by the (mesh-based) underlying ground truth, and I am not sure that it makes sense to learn arbitrary resolution implicit representation on top of it - for this method to be more meaningful, I believe there is a need on either adding ability to estimate geometry from images (as a supervision, not just conditioning), or e.g. from multi-view reconstructions.
> > > > Thus I stand by my original rating.

---

> > > > > ### Author Response · Authors · 2023-08-19
> > > > >
> > > > > Thank you for your comment. However, we would like to reiterate that our primary goal was not to learn high-frequency shape details “on top of” the ground truth mesh resolution, but to estimate the hand shapes themselves only from images. We showed that our method reconstructs shapes that are closer to the ground truth shapes (even though the ground truth is given as MANO meshes) than the actual MANO-based methods [19, 44, 45] (Table 1 in the paper) by leveraging denser pixel-aligned features. **Our motivation for using implicit representation directly follows the motivation of the existing methods [15, 16, 18, 24, 25, 29, 39] that also used the ground truth MANO [34] or SMPL [21] meshes for training implicit functions**. Therefore, we would greatly appreciate it if you could reconsider your rating.
> > > > >
> > > > > Once again, we thank you for engaging in discussions with us.

---

### Official Review · Reviewer_Uqr4 · 2023-07-06

**Soundness:** 3 good
**Presentation:** 3 good
**Contribution:** 3 good
**Rating:** 6
**Confidence:** 3

**Summary:**

This paper introduces an implicit spatio-temporally continuous hand representation for RGB videos. Firstly, based on LEAP [25], the occupancy function and LBS weights are pretrained as priors for query points. Then two query flow representations are introduced to model the skeleton and the shape, respectively. The query flow representations are Fourier coefficients, which can be treated as a low-pass filter for 4D representations. Experiments on InterHand2.6M and RGB2Hands show the proposed method achieves accurate and efficient 4D predictions.

**Strengths:**

- The proposed 4D implicit representation is highly efficient during both training and testing.
- The Fourier coefficients are effective to get smooth and continuous temporal dynamics.

**Weaknesses:**

- [Balance] This paper proposes using Fourier coefficients to model 4D hand and emphasizes this will guarantee smooth and continuous temporal dynamics. Especially, it learns coefficients for N=6 basis functions (Line 175). I am wondering how Fourier coefficients and N balance the accuracy, the efficiency and the smoothness. For example, will N=6 lead to over smooth? why not use a larger N? is it because of efficiency?

- [Optimisation] Unlike existing representations, the proposed representation is coefficients. I am wondering if this will raise any training or optimation issues. For example, will this representation require more epochs to converge or be harder to fit during training?

- [$\Psi$] For pose flow, there is an off-the-shelf pose estimator $\Psi$. I'm curious about the role this pose estimator plays. Is it like a condition that directly determine the final performance or an initial values that only reduce search space (Line 209) or a refinement (Tab.3)? Would it possible to use different $\Psi$ with varied performance and see the impact on the final results during training? Also, what will happen if $\Psi$ provides bad predictions during testing? What if we do not have $\Psi$?

- [Metric] It would be better to compare SOTA methods like [19,44,45] wrt other metrics like MPJPE, even if they focus on a single frame or are based on MANO.

**Questions:**

See weaknesses.

**Limitations:**

Yeah.

---

> ### Author Rebuttal · Authors · 2023-08-06
>
> We thank the reviewers for their thoughtful feedback and finding that the proposed method is “novel and interesting” [vgcD] and shows “large improvements” [rxcm, g8GH]. We also appreciate [rxcm] for finding that the “paper is clearly written and well-motivated.” In what follows, we address the concerns of the reviewers.
>
> ---
>
> > **Weakness 1.** “[Balance] This paper proposes using Fourier coefficients to model 4D hand and emphasizes this will guarantee smooth and continuous temporal dynamics. [...] why not use a larger N? is it because of efficiency?”
> >
>
> **Reply:** In Table S1 in the supplementary, we provide our 4D hand reconstruction results with the varying value of $N =$ {$4, 6, 8$}. As discussed in lines 19-21 in the supplementary, our model performance is slightly decreased when $N$ is too small ($N = 4$), while it is not affected much by $N$ when $N ≥ 6$. Thus, we choose $N = 6$ in our main method to achieve a good balance between accuracy and computational efficiency. We did not consider a value beyond 8 for $N$, as it exceeds the Nyquist frequency given $T = 17$.
>
> > **Weakness 2.** “[Optimisation] Unlike existing representations, the proposed representation is coefficients. I am wondering if this will raise any training or optimization issues. For example, will this representation require more epochs to converge or be harder to fit during training?”
> >
>
> **Reply:** Although our model prediction is done in the coefficients space, the whole forward process is differentiable (note that Eq. (1) is defined by the linear combination of sine and cosine functions), and we have observed no particular difficulty during the model optimization. About the training convergence, in Figure R2 (in PDF), we show the validation loss graphs of our model and the variation of our model based on the direct shape-space prediction (*Fourier → InputCond* in Table 4 in the paper). There is no significant difference in the training convergence trend, and both models converge approximately after 100K steps in training.
>
> > **Weakness 3.** “[Ψ] For pose flow, there is an off-the-shelf pose estimator Ψ. I'm curious about the role this pose estimator plays. Is it like a condition that directly determine the final performance or an initial value that only reduce search space (Line 209) or a refinement (Tab.3)? Would it possible to use different Ψ with varied performance and see the impact on the final results during training? Also, what will happen if Ψ provides bad predictions during testing? What if we do not have Ψ?”
> >
>
> **Reply:** $\Psi$ provides the initial (noisy) pose values to reduce the search space of our model prediction, and here we demonstrate that our final shape accuracy is quite robust to the quality of the initial pose predictions. In the table below, we evaluate our 4D reconstruction results using two different models for $\Psi$ [19, R1]. While the initial pose accuracies have a considerably large gap (3.7mm in MPJPE), our final shape accuracies do not vary much, and **both settings achieve the state-of-the-art 4D reconstruction results** compared to the baseline methods in Table 1 in the paper.
>
> | $\Psi$ | Initial joint error (MPJPE $\downarrow$) | Shape accuracy (IoU $\uparrow$) | Shape accuracy (CD $\downarrow$) |
> | --- | --- | --- | --- |
> | **IntagHand [19]** | 13.3 | 62.8 | 4.46 |
> | **DIGIT [R1]** | 17.0 | 62.2 | 4.58 |
>
> [R1] Fan *et al.*, Learning to disambiguate strongly interacting hands via probabilistic per-pixel part segmentation. In 3DV, 2021.
>
> To additionally examine “*what will happen if $\Psi$ provides bad predictions during testing”*, we also measured our 4D reconstruction accuracy after injecting uniformly sampled noises with the amplitude [−$x$, +$x$] to every dimension of the initial pose prediction in the test time — following the exact ablation study setting in [16] (Section D in the supplementary). As shown in the table below, our model performance is quite robust to the injected noises (gently decreasing its accuracy for a higher level of noise), and again, **all settings achieve the state-of-the-art results** compared to the baselines in Table 1 in the paper.
>
> | Noise level ($x$)  | Shape accuracy (IoU $\uparrow$) | Shape accuracy (CD $\downarrow$) |
> | --- | --- | --- |
> | **0mm (No noise)** | 62.8 | 4.46 |
> | **1mm** | 62.7 | 4.48 |
> | **3mm** | 62.5 | 4.50 |
> | **5mm** | 62.2 | 4.53 |
>
> Currently, our method does not assume a situation where $\Psi$ does not exist. We will further investigate this direction in our future work.
>
> > **Weakness 4.** “[Metric] It would be better to compare SOTA methods like [19,44,45] wrt other metrics like MPJPE, even if they focus on a single frame or are based on MANO.”
> >
>
> **Reply:** Thank you for your suggestion. Although our primary goal of this research was accurate shape reconstruction rather than joint estimation, we additionally evaluated our results on MPJPE in comparison to [19, 44, 45], as shown in the table below. As [19] and [45] originally use the additional *ground truth bone length* for rescaling the joint predictions, we use the mean bone length of the hands in the training set of InterHand2.6M [29] following [18] to perform fair comparisons. Our method achieves a lower (better) MPJPE than the compared methods on InterHand2.6M dataset.
>
> | $\Psi$ | Joint error (MPJPE $\downarrow$) |
> | --- | --- |
> | **ACR\*[44]** | 18.3 |
> | **Two-Hand-Shape-Pose [45]** | 15.9 |
> | **IntagHand [19]** | 13.3 |
> | **Ours** | 11.0 |

---

> > ### Comment · Reviewer_Uqr4 · 2023-08-19
> >
> > Thanks for the authors' rebuttal. I am satisfied with the authors' responses regarding [$\Psi$] and [Metric]. However, I still do not fully understand for other two. At this time, I still keep my rating.
> >
> > [Balance] The authors claim their selection is a good balance.  However, Table S1 only highlights the accuracy, not the computational efficiency. Would it be possible to provide computational efficiency like Tab. 2? Also,  the author did not reply to me about the temporal over-smooth issue. But I find the reply from other reviewers part, I think it is OK. As three of the reviewers are concerned about the over-smooth issue, I think this part could be highlighted in the revision.
> >
> > [Optimisation] I do not agree with the answer to this part. Optimization is related to convergence speed and convergence accuracy. Based on Figure R2, it seems (a) is faster with a higher loss than (b). Does this imply coefficient learning lead to a slower speed and better accuracy? why?

---

> > > ### Author Response · Authors · 2023-08-19
> > >
> > > Thank you for your thoughtful comment. We additionally provide our response to your questions below.
> > >
> > > **[Balance]** In the table below, we additionally show the computational efficiency with the varying number of $N$ (along with the accuracy reported in Table S1) to further support our claim that $N=6$ achieves a good balance between accuracy and computational efficiency. We will add these additional results in the revision.
> > >
> > > |  | **IoU (%)** $\uparrow$  | **CD (mm)** $\downarrow$ | **L1-Corr (mm)** $\downarrow$ | **Training Time (sec.)** $\downarrow$ | **Inference Time (sec.)** $\downarrow$ |
> > > | --- | --- | --- | --- | --- | --- |
> > > | **N=4** | 62.5 | 4.50 | 10.9 | **2.50** | **0.12** |
> > > | **N=6** | **62.8** | 4.46 | **10.8** | 2.75 | 0.15 |
> > > | **N=8** | 62.6 | **4.45** | **10.8** | 2.87 | 0.18 |
> > >
> > > Following your suggestion, we will also make sure to highlight our discussion on the “temporal over-smooth” issue in the revision.
> > >
> > > **[Optimisation]** We still think that there is no *significant* difference in the convergence speed between (a) and (b), and both reach the validation loss of $3.3 \times 10^{-3}$ approximately after 100K steps. The validation loss of (b) *slightly* decreases further after 100K steps -- leading to a better accuracy compared to (a) (Table 4 in the paper) as you mentioned. We believe that this is because our flow estimation in the frequency-domain subspace naturally guarantees smooth temporal deformations, which better match the characteristics of our target motion to reconstruct (i.e., hand motions captured in InterHand2.6M [26]) — as also discussed in our response to Question 1 of [vgcD].
> > >
> > > If our response and the additional experimental results have addressed most of your concerns (including **[$\Phi$]** and **[Metric]** addressed in our previous response), we would greatly appreciate it if you kindly consider updating your rating. Once again, we thank you for your insightful discussions.

---

> > > > ### Comment · Reviewer_Uqr4 · 2023-08-19
> > > >
> > > > Thanks for the discussion. I think most of my concern has been solved.  I will raise my final rating.

---

### Official Review · Reviewer_rxcm · 2023-07-06

**Soundness:** 4 excellent
**Presentation:** 4 excellent
**Contribution:** 4 excellent
**Rating:** 7
**Confidence:** 5

**Summary:**

This paper proposes FourierHandFlow, a 4d hand pose and shape representation that inherently uses Fourier series as query flow representation. Given RGB sequence, a fixed number of Fourier series are learned to represent hand pose and shape. The authors use two types of flows to decompose pose and shape: pose flow (joint flow) and shape flow. Such decomposition makes it more efficient to reconstruct 4d hands. Experiments on Interhand2.6M and RGB2Hands datasets demonstrate its superiority over existing two hand estimation methods.

**Strengths:**

This paper is clearly written and well-motivated.  The compact Fourier series representation is proved to be effective for two hand 4d reconstruction in a smoother way. The video results are impressive. The quantitative results show large improvements.

**Weaknesses:**

The main weakness is that, I am not very sure whether such representation could correctly model the surface details (or pose dependent deformation) of the hand because Fourier series based representation seem to generate over smoothed results due to its low dimension nature of shape flow. For example, at video 04:06 (frame 7101), the little finger of the left hand in “Alt. View 1” is too thin.

**Questions:**

1. How to obtain the gt 3d shape of InterHand2.6M for training? Is that simply the MANO labeling? If true, what’s the advantage of using implicit representation instead of MANO for hand shape learning?

2. L. 228, what does ^2 mean? Is that a mistake?


**Limitations:**

Yes.

---

> ### Author Rebuttal · Authors · 2023-08-06
>
> We thank the reviewers for their thoughtful feedback and finding that the proposed method is “novel and interesting” [vgcD] and shows “large improvements” [rxcm, g8GH]. We also appreciate [rxcm] for finding that the “paper is clearly written and well-motivated.” In what follows, we address the concerns of the reviewers.
>
> ---
>
> > **Weakness 1.** “I am not very sure whether such representation could correctly model the surface details (or pose dependent deformation) of the hand because Fourier series based representation seem to generate over smoothed results due to its low dimension nature of shape flow.”
> >
>
> **Reply:** We want to clarify that we used Fourier series **along the temporal** **axis** (Eq. (1) in the paper) to model the smooth temporal change of hand shapes for 4D reconstruction. Although such representation may cause over-smoothed shape change **along the temporal (*t*) axis**, our shape prediction **along the spatial (*x, y, z*) axes** is not bounded by the frequency-domain subspace because we did not apply Fourier series along the spatial axes unlike in the existing methods [9, 13, 16]. We will make sure to clarify this point in the revision.
>
> > **Question 1.** “How to obtain the gt 3d shape of InterHand2.6M for training? Is that simply the MANO labeling? If true, what’s the advantage of using implicit representation instead of MANO for hand shape learning?”
> >
>
> **Reply:** The ground truth shapes of InterHand2.6M [26] are given as meshes with MANO [34] topology. Following the 3D/4D reconstruction methods [15, 16, 18, 24, 25, 29, 39] that similarly use the ground truth meshes with a fixed topology (e.g. MANO [34] or SMPL [21] meshes) for training implicit functions, we believe that using implicit representation introduces several advantages over mesh-based representation. Most importantly, implicit representation utilizes the pixel-aligned features directly corresponding to the **dense** query points, often leading to more accurate shape reconstructions that are better aligned to the input image [9, 18]. In Table 1 in the paper, we experimentally demonstrated that our method reconstructs more accurate shapes than the existing methods based on MANO parameter [44, 45] or MANO vertex regression [19] on InterHand2.6M. We also note that, as implicit representations learn continuous shape field, they can naturally reconstruct shapes in an arbitrary resolution (including the extrapolated resolution beyond the observed ground truth resolution). In Figure R1 (in PDF), we show the qualitative reconstruction examples of the current state-of-the-art mesh-based reconstruction method [19] and ours on InterHand2.6M dataset, where ours produces more plausible hand shapes. We will add this discussion in the revision.
>
> > Question 2. *“L. 228, what does ^2 mean? Is that a mistake?”*
> >
>
> **Reply:** ^2 denotes *footnote 2*. We will change the footnote symbol for better clarity in the revision. Thank you for pointing this out.

---

> > ### Comment · Reviewer_rxcm · 2023-08-11
> > **reply**
> >
> > I have read the rebuttal. The authors solved my concerns.

---

### Official Review · Reviewer_g8GH · 2023-07-07

**Soundness:** 2 fair
**Presentation:** 3 good
**Contribution:** 3 good
**Rating:** 5
**Confidence:** 4

**Summary:**

This paper introduces FOURIERHANDFLOW, which is a spatio-temporal continuous representation for the human hands. It combines a continuous 3D hand occupancy field with articulation-aware query flows represented as Fourier series along the temporal axis. These query flows are parameterized by coefficients learned from an input RGB sequence. Specifically, two types of Fourier query flows, namely pose flow and shape flow, are used to address the challenges of continuous and smooth 4D reconstruction, computational efficiency, and articulated shape modeling with correspondences.

**Strengths:**

- This paper provides a well-articulated description of the problems with existing implicit methods. It presents a clear logical flow and addresses specific challenges effectively. The results show particularly noticeable improvements for abrupt or jittery motions correction.
- Although Fourier series is not a novel concept, this paper extends it to the temporal dimension.

**Weaknesses:**

- It is undeniable that this work is closely related to the Fourier Occupancy Field [9]. The authors should consider adding comparative experiments with this work.
- This paper focuses on describing the acquisition of Fourier Query Flow in the method section but lacks a direct description of how the final pipeline for generating the 4D hand is constructed (e.g., how the pre-trained canonical field occupancy is utilized). A more comprehensive overview of this aspect should ideally be included in the first paragraph of the methodology section for better clarity.
- The pose flow and the shape flow are two important branches proposed in this paper, and the final flow is the sum of both. It would be helpful to provide clearer illustrations that show the trajectories generated by each branch separately, as well as the combined trajectory. This would enable readers to gain a clearer understanding of the method.

**Questions:**

- In the prediction of pose flow, I think that the essence of being able to perform articulated shape modeling is to represent the skeleton of the hand in the form of a graph, which can maintain the topological structure of the hand. Therefore, in this process, the overall skeleton of the hand is predicted frame by frame. But the subsequent prediction of Fourier coefficients is based on the trajectory of each joint, so how is the temporal smoothness of each joint constrained here?
- For different input sequences, is it necessary to normalize them to the same sampling space? If so, what are the specific implementation details?
- In the pose flow, the paper estimates the joint flow of the keypoints and then propagates it to arbitrary points based on LBS. Therefore, the estimation of the shape flow after arbitrary points heavily will rely on the accuracy of LBS. Does the paper have any special handling or explanation for this situation?
- Recommendation: For the ODE based method, although its computational complexity is relatively high, it can maintain the property of Diffeomorphism. This aspect can be considered later.

**Limitations:**

The authors have partially addressed the limitations of their work, though there is space for improvement (see the section Strengths And Weaknesses).

---

> ### Author Rebuttal · Authors · 2023-08-06
>
> We thank the reviewers for their thoughtful feedback and finding that the proposed method is “novel and interesting” [vgcD] and shows “large improvements” [rxcm, g8GH]. We also appreciate [rxcm] for finding that the “paper is clearly written and well-motivated.” In what follows, we address the concerns of the reviewers.
>
> ---
>
> > **Weakness 1.** “It is undeniable that this work is closely related to the Fourier Occupancy Field [9]. The authors should consider adding comparative experiments with this work.”
> >
>
> **Reply:** Fourier Occpancy Field (FOF) [9] uses Fourier Series along one of the spatial axes (i.e., z-axis) to enable efficient 3D human reconstruction, while our method uses Fourier Series along the temporal axis (1) to regularize high-frequency (e.g., jittery) temporal shape change and (2) to enable efficient 4D hand reconstruction. Along with this difference, we believe that another important difference between FOF and our method lies in the characteristics of the shape category that each aims to reconstruct. FOF aims to reconstruct clothed humans, which have complex shape variations (e.g., cloth wrinkles) but with fewer self-occlusions caused by the underlying shape articulations. In contrast, our method aims to reconstruct human hands, which have more complex articulations leading to more severe self-occlusions. Thus, most of the existing hand implicit functions, e.g., [6, 16, 18], use articulated implicit representation to directly model pose-dependent deformations with respect to the learned canonical hand shape to incorporate strong pose prior. Similarly, one of our main contributions was also to propose articulation-aware query flows (i.e., pose and shape flows) to directly model pose-dependent 4D hand deformations. Since FOF is a non-articulated implicit function, we thought that the existing state-of-the-art methods specifically designed for the same hand reconstruction task would be more challenging baselines. In the table below, we show the experimental comparisons with FOF, where FOF yields similar results to our non-articulated implicit function baseline (Occupancy Flow [29]) in Table 1 in the paper. We will add this discussion and the experimental results in the revision.
>
> | | Shape accuracy (IoU $\uparrow$) | Shape accuracy (CD $\downarrow$) |
> | --- | --- | --- |
> | FOF [9] | 30.4 | 23.88 |
> | Ours | 62.8 | 4.46 |
>
>
> >**Weakness 2.** “A more comprehensive overview of this aspect should ideally be included in the first paragraph of the methodology section for better clarity.”
> >
>
> > **Weakness 3.** “It would be helpful to provide clearer illustrations that show the trajectories generated by each branch separately, as well as the combined trajectory. This would enable readers to gain a clearer understanding of the method.”
> >
>
> **Reply:** Thank you for your suggestion. We will make sure to (1) add the overview of our overall pipeline and (2) show the trajectories generated by each type of flow in the revision.
>
> >**Question 1.** “In the prediction of pose flow, I think that the essence of being able to perform articulated shape modeling is to represent the skeleton of the hand in the form of a graph, which can maintain the topological structure of the hand. [...] But the subsequent prediction of Fourier coefficients is based on the trajectory of each joint, so how is the temporal smoothness of each joint constrained here?”
> >
>
> **Reply:** The temporal smoothness of each joint is preserved from our trajectory prediction in the *frequency-domain subspace* (lines 49-50). For maintaining the topological structure in our temporal reconstruction, we feed the joint features extracted using a graph convolutional network (GCN) as inputs to our Fourier coefficients estimation module (lines 202-203). We observe that such structure-aware input feature conditioning is sufficient to be able to preserve the hand topological structure in our temporal reconstruction as shown in the supplementary video.
>
> >**Question 2.** “For different input sequences, is it necessary to normalize them to the same sampling space? If so, what are the specific implementation details?”
> >
>
> **Reply:** Along the spatial dimension, we normalized the 3D coordinate space of each input sequence by aligning the predicted hand root joint of the first frame to the origin point. Along the temporal dimension, we did not apply normalization. We will clarify this in the revision.
>
> >**Question 3.** “[...] the estimation of the shape flow after arbitrary points heavily will rely on the accuracy of LBS. Does the paper have any special handling or explanation for this situation?”
> >
>
> **Reply:** The accuracy of the estimated shapes after LBS (S.3.1 in the supplementary paper) is dependent on two factors: (1) the accuracy of the pre-trained skinning weights $\mathbf{w}^{\mathbf{p}}$ and (2) the accuracy of our joint flow estimation, which determines the rigid bone transformations {$\mathbf{T}_b$} for $b=1, \textit{...}, B$. To examine factor (1), we calculated the accuracy of the pre-trained $\mathbf{w}^{\mathbf{p}}$ by computing the shape IoU after applying LBS using the ground truth rigid bone transformations. As the resulting shape IoU was quite high (98.62% on InterHand2.6M [26] test dataset), our shape accuracy after LBS would mainly depend on the remaining factor (2), which is the accuracy of our joint flow estimation. Thus, we performed experiments to examine the robustness of our final shape accuracy with varying quality of the initial joint estimation and showed that our method is quite robust to it. We kindly refer you to our answer to Weakness 3 of the reviewer [Uqr4] for the experimental results.
>
> >**Question 4.** “Recommendation: For the ODE based method, although its computational complexity is relatively high, it can maintain the property of Diffeomorphism. This aspect can be considered later.”
> >
>
> **Reply:** Thank you very much for your valuable comment. We will consider this aspect in our future work.

---

### Author Rebuttal · Authors · 2023-08-08

We provide the figures referred to in our author response in the PDF file below.

---

### Decision · Program_Chairs · 2023-09-21

**Decision:**

Accept (poster)

**Comment:**

This paper got mostly positive ratings  (A, WA, WA, BA, R) from reviewers.

The major strength of this paper lies in their spatio-temporally continuous implicit representation based on as Fourier series, supported by the performance improvements shown in their experiment.

The major concern particularly raised by the reviewer on the negative side is regarding the less clear presentations on some parts and the motivation of the approach in terms of applying implicit representation on the relatively spare MANO mesh GTs.

The authors resolved most concerns by other reviewers in their rebuttal and discussion phase.

The AC is inclined to the positive side, given the novel part in their representation and experimental demonstrations. The AC also found that the authors convincingly answered the questions and concerns raised by reviewers. Overall, the paper meets the bar of NeurIPS.
Authors should very carefully check the reviewer’s feedback and improve the quality of the paper for the camera-ready version.